# How to Combine Solar Geoengineering and Mitigation under Climate Targets?

Mohammad M. Khabbazan[1,2,3,4], Marius Stankoweit[1], Elnaz Roshan[1], Hauke Schmidt[5], and Hermann Held[1,4]

[1]Research Unit Sustainability and Global Change (FNU), University of Hamburg, Grindelberg 5, 20144 Hamburg, Germany.

[2]Workgroup for Economic and Infrastructure Policy (WIP), The Technical University of Berlin, Strasse des 17. Juni 135, 10623 Berlin, Germany.

[3]Department of Energy, Transport, and Environment (EVU), The German Institute for Economic Research (DIW Berlin), Mohrenstrasse 58, 10117 Berlin, Germany.

[4]Center for Earth System Research and Sustainability (CEN), University of Hamburg, Bundesstr. 53, 20146 Hamburg, Germany.

[5]Max Planck Institute for Meteorology, Bundesstr. 53, 20146 Hamburg, Germany.

*Correspondence to:* Mohammad M. Khabbazan (mohammad.khabbazan@uni-hamburg.de)

**Abstract.** So far, scientific analyses have mainly focused on the pros and cons of solar geoengineering or solar radiation management (SRM) as a climate policy option in mere isolation. Here, we put SRM into the context of mitigation by a strictly temperature-target-based approach. As the main innovation, we present a scheme that extends the applicability regime of temperature targets from mitigation-only to SRM-mitigation analyses. We explicitly account for one major category of side effects of SRM while minimizing economic costs for complying with the 2°C temperature target. To do so, we suggest regional precipitation guardrails that are compatible with the 2°C target. Our analysis shows that the value system enshrined in the 2°C target leads to an elimination of most of SRM from the policy scenario if a transgression of environmental targets is confined to 1/10 of the standard deviation of natural variability. Correspondingly, about half to nearly two-thirds of mitigation costs could be saved, depending on the relaxation of the precipitation criterion. In addition, assuming a climate sensitivity of 3°C or more, in case of a delayed enough policy, a modest admixture of SRM to the policy portfolio might provide debatable trade-offs compared to a mitigation-only future. Also, in our analysis which abstains from an utilization of negative emissions technologies, for climate sensitivities higher than 4°C, SRM will be an unavoidable policy tool to comply with the temperature targets. The economic numbers we present must be interpreted as upper bounds in the sense that cost-lowering effects by including negative emissions technologies are absent. However, with an additional climate policy option such as carbon dioxide removal present, the role of SRM would be even more limited. Hence, our results, pointing to a limited role of SRM in a situation of immediate implementation of a climate policy, are robust in that regard. This limitation would be enhanced if further side effects of SRM are taken into account in a target-based integrated assessment of SRM.

# 1 Introduction

Since Paul Crutzen has highlighted solar radiation management (SRM) as a potential climate policy option in addition to adaptation and mitigation (Crutzen (2006)), there is increasing research on this technique as a measure to counteract anthropogenic global warming (Barrett et al. (2014); Bellamy et al. (2013); Goes et al. (2011); Irvine et al. (2012); Kravitz et al. (2013); MacMartin et al. (2014); Moreno-Cruz and Keith (2013); Schmidt et al. (2012); Shepherd (2009); Wigley (2006)). The bulk of analyses focuses on the pros and cons of SRM in mere isolation. However, an integrated analyses is needed to allow decision-making on SRM taking into account more conventional policy options such as adaption or mitigation.

In a non-welfare-optimal setting, Smith and Rasch (2013) studied the role of SRM in conjunction with mitigation for a limited set of pre-defined mitigation scenarios inspired by the Representative Concentration Pathways (RCP) to meet a pre-defined temperature target. A few studies have performed an integrative analysis comprising both SRM and a stylized representation of mitigation in a Cost-Benefit Approach (CBA) which is arguably the most prominent welfare-optimal approach (Bahn et al. (2015); Emmerling and Tavoni (2018); Goes et al. (2011); Heutel et al. (2016); Heutel et al. (2018); Moreno-Cruz and Keith (2013)). However, because the economic costs of SRM have been assumed to be low compared to mitigation, any assessment should consider the inclusion of side effects of SRM. Earlier studies presented trade-off results for stylized impact assumptions within the standard economic paradigm of CBA which is, as much as possible, in line with standard economic axioms. Nevertheless, some studies suggest that directly recommending climate policy through only CBA is challenging due to the presence of deep uncertainty about global warming impact functions (Ekholm (2018); Kolstad et al. (2014); Kunreuther et al. (2014)). These studies suggest using a target-based approach, known as Cost-Effectiveness Analysis (CEA), as long as no better data is available (Kunreuther et al. (2014); Neubersch et al. (2014)).

Furthermore, for pragmatic reasons, one might argue that analyses should reflect the consequences of climate targets. Along that line, Lawrence et al. (2018) put climate engineering proposals into the context of climate targets, however, without performing CEAs. In addition, while Arino et al. (2016), Ekholm and Korhonen (2016), and Emmerling and Tavoni (2018) evaluated SRM together with mitigation applying CEA, an inclusion of side effects of SRM was not in their focus. In particular, these studies did not define clear guardrails for side-effects of SRM.

To the best of our knowledge, for the first time, we introduce and apply a concept for an integrated analysis of SRM and mitigation in line with global mean temperature targets which also integrates one side effect of SRM.[1] The 2°C target is the cornerstone of the Paris agreement (UNFCCC, 2015 [UNFCCC (2015): Adoption of the Paris Agreement. FCCC/CP/2015/L.9/Rev.1.]). It encapsulates society's aversion against deeply uncertain global warming impacts (Neubersch et al. (2014); Schellnhuber (2010)). Driven by the expectation that costs of transforming the energy system can be projected much more robustly than the aggregate impacts of global warming (Stern (2007)), a plethora of economic mitigation analyses has derived cost-minimal energy scenarios, which comply with this target (Edenhofer et al. (2014)).

---

[1]Based on the previous version of this article (Stankoweit et al. (2015)), Roshan et al. (2019) applied a Cost-Risk Analysis and evaluated the optimal SRM in conjunction with mitigation, considering regional disparities in the precipitation risks.

However, if SRM is employed, the global mean temperature is no longer a good proxy for regional climate impacts because SRM causes patterns of regional climate change that would differ from those induced by greenhouse gas forcing (Kravitz et al. (2013); Oschlies et al. (2017)). This particularly applies to regional precipitation changes (Shepherd (2009); Bala et al. (2008); Robock et al. (2008)). Accordingly, and as a key innovation of this article, we suggest extending the regime of applicability of the 2°C target from mitigation-only to joint SRM-mitigation portfolios when global mean temperature and regional precipitation are simultaneously considered. The next subsection will describe how we generalize the global mean temperature-target concept to consider such regional climate effects induced by SRM. For our joint SRM-mitigation analysis, we utilize the integrated energy-economy-climate model MIND (Edenhofer et al. (2005)), which provides one of the simplest possible options to distinguish the renewable sector from the fossil-fuel sector under induced technological change. We extend the model to include a spatially explicit resolution in terms of 'Giorgi regions' (Giorgi and Bi (2005)) and run specific policy scenarios showing the trade-offs between mitigation and SRM. We also highlight the most important factors that derive our results.

The rest of the paper is organized as follows. Section 2 provides details of the innovated guardrails, data, and the numerical model employed. Section 3 presents the results. Some sensitivity analyses are presented in Section 4, and Section 5 concludes the paper.

## 2   Methods

### 2.1   Precipitation guardrails

The application of SRM to counteract greenhouse gas-induced global warming results in regional terrestrial precipitation patterns. These patterns differ from the pure greenhouse-gas-induced ones (Shepherd (2009); Bala et al. (2008); Robock et al. (2008)). Hence, when the climate system is forced by greenhouse gases and SRM simultaneously, global mean temperature ceases to be a good proxy for regional precipitations. Accordingly, to preserve the target concept, one must bypass the destroyed link between global mean temperature and regional climate. Hence, modeling regional climate explicitly and inventing respective regional targets are needed.

In the following, a set of regional targets is defined in a distinct way such that it preserves the meaning of the global mean temperature target concept. Thereby, we add a necessary condition to respect the targets only implicitly included in the orginal 2°C target. Here we ask: 'For a given region, what would be its climate anomaly in a 2°C warmer world without SRM?' This regional climate anomaly is the maximum climate change acceptable for a decision-maker who accepts global warming of 2°C. (In the target-based literature, 'acceptable' for further consideration is called 'admissible' (Bruckner et al. (2008); Kriegler and Bruckner (2004); Petschel-Held et al. (1999)). Hence, we confine regional climate change to the intervals of climate variables that would be spanned by ramping the global mean temperature anomaly up from zero to 2°C. We augment the 2°C target by this exact set of intervals as the more fundamental target. We suggest to generate such intervals because the 2°C target has emerged from a line of argument excluding SRM (Schellnhuber (2010)). This region-based, hence more fundamental, target would then be valid also for technology portfolios which include SRM. Analogous to the original global target, this target also bypasses the criticized monetarization of climate impacts on which CBAs is based.

Here, we focus on a subset of regional climate guardrails in terms of precipitation changes becaue these have been highlighted as a critical drawback of SRM. While regional temperature (Asseng et al. (2011)) and precipitation (Portmann et al. (2010)) are highly relevant for agricultural productivity, the 'pattern mismatch' (i.e. the discrepancy between greenhouse gas- and SRM-induced patterns) of precipitation is of a larger order of magnitude than that of temperature (Kravitz et al. (2014)). We are not claiming that temperature and precipitation are the only relevant climate predictors for agricultural productivity or the functionality of ecosystems in general. Still, we acknowledge precipitation limits as sensible boundary conditions within a target-based framework.

Figure 1 shows our suggested guardrails for two hypothetical regions $r_1$ and $r_2$. For this figure, as well as our whole analysis, we employ the following assumptions: (i) Regional climate anomalies can be approximated as a superposition of anomalies induced by greenhouse gases and by SRM, respectively (Ban-Weiss and Caldeira (2010)), (ii) both regional components scale linearly with their corresponding global mean temperature component (Frieler et al. (2012); Ricke et al. (2010)).

Equations 1 and 2 formalize our suggested guardrails for the admissible precipitation anomalies ($\Delta P_R$) for all regions ($R$):

$$\forall_{r_1 \in R} \text{ with } C(r_1, \text{CO}_2) > 0 : \ -EA \leq \Delta P_{r_1} \leq \Delta P_{r_1}^{2°C} + EA \tag{1}$$

$$\forall_{r_2 \in R} \text{ with } C(r_2, \text{CO}_2) < 0 : \ \Delta P_{r_2}^{2°C} - EA \leq \Delta P_{r_2} \leq EA \tag{2}$$

where $\Delta P_R^{2°C}$ denotes the regional precipitation anomoly of a 2°C warmer world without SRM.

$r_1$ is characterized by a positive $\text{CO}_2$ (Greenhouse-gas-driven) scaling coefficient ($C(r_1, \text{CO}_2) > 0$), which denotes a positive change in precipitation ($P$) when the global mean temperature ($T$) rises. $r_2$ is, however, characterized by a negative $\text{CO}_2$ scaling coefficient ($C(r_2, \text{CO}_2) < 0$). The green bands in panel a) and panel b) define the regular admissible area for precipitation change compatible with the 2°C target. As noted earlier, SRM imprints patterns of regional precipitation and temperature change that would differ from those induced by greenhouse gas forcing (Kravitz et al. (2013)). Therefore, the regular admissible range of those regions where SRM scaling coefficients have the same sign as their $\text{CO}_2$ scaling coefficients would prohibit any SRM use. To avoid this, an extra range of admissibility ($EA>0$) is required. We pragmatically suggest adding a fraction of regional standard deviation, derived from inter-annual variability, on both ends of the admissibility range. These extra ranges are depicted in blue. In this paper, we consider 5% and 10% of the standard deviation of inter-annual variability. In Sect. 4.1, we analyze the sensitivity of our results to the size of these ranges. While this extra admissible range, as our whole analysis, is not based on formalized impacts, it is ethically and formally consistent with the assumptions of CEA.

## 2.2 Regional scaling coefficients and natural variability

For the regionalization of climate change effects, we use 'Giorgi regions' (Giorgi and Bi (2005)) (see Table 1 and Fig. 2) which are, very roughly, consistent with synoptic scales. This regional resolution brings about a markedly different image from a global average in the sign and magnitude of effects in the climatic variables under scrutiny. At the same time, it avoids a larger

number of simultaneous regional targets that might be perceived as too restrictive.[2] However, we stress that the choice of the resolution is ultimately a normative decision to be taken by society.

For the scaling coefficients, we diagnose annual mean regional precipitation changes from linear pattern scaling (Ricke et al. (2010)) which are derived as a linear superposition (Ban-Weiss and Caldeira (2010)) of greenhouse-gas-induced and SRM-induced changes in global mean temperature. We use the outputs of nine atmosphere-ocean general circulation models (AOGCMs).[3] The average greenhouse-gas-induced scaling coefficients including their sample standard deviations ($c_{CO_2}$[%/K] and $\sigma_{c_{CO_2}}$) and the average SRM-induced scaling coefficients including their sample standard deviations ($c_{SRM}$ [%/K] and $\sigma_{SRM}$) from the nine AOGCMs are shown in Table 1. Figure A1 in the Appendix additionally shows the variations of scaling coefficients for each region from nine AOGCMs. The scaling coefficients may switch the sign for some regions if a specific AOGCM is considered or not. Table 1 also shows the ratio $R_r = c_{SRM}/c_{CO_2}$, which is used to indicate co-effects of SRM and $CO_2$ that link temperature effects to precipitation effects. All regions are characterized by SRM and $CO_2$ coefficients that increase or decrease precipitation in opposite directions.Hence, $R_r$ is negative in all regions. In regions where $0 > R_r > -1$, SRM under-compensates $CO_2$-induced precipitation changes. However, in regions where $-1 > R_r$, SRM over-compensates $CO_2$-induced precipitation changes.

An important note is that, from the average scaling coefficients, their sample standard deviation, and the assumption of a normal distribution of scaling coefficients, we can determine the probability that the signs of scaling coefficients in some regions switch. For example, $c_{CO_2}$ and $c_{SRM}$ in Amazonia are -1.35 and 0.18 %/K, respectively. Yet, the corresponding sample standard deviations are 2.39 and 2.43 %/K, which are large enough to likely switch the sign of scaling coefficients when scaling coefficients are randomly generated with normal distribution. Therefore, it is also likely that $R_r$ becomes positive for some regions, which means SRM and $CO_2$ both either increase or decrease precipitation. In Sect. 4.2, we conduct a Monte Carlo analysis by randomly choosing the scaling coefficients from the above distributions.

We determine the standard deviation of natural variability in precipitation the following way. We use three data sets of annual precipitation, aggregated to Giorgi regions based on GPCC_WATCH (1901-2001) (Weedon et al. (2010)), PGFV2 (1901-2012) (Sheffield et al. (2006)), and GPCC_WFDEI (1979-2010) (Weedon et al. (2014)). Then, for any region $r$ and data set $s$, we determine the precipitation means $\mu_{r,s}$. To obtain the standard deviation of natural variability as distinct from global warming, for any $r$, $s$, we first subtract a polynomial fit of second order of the time evolution from the time series for detrending. The detrended data represent a more significant variability linked with a distinct time scale of the data (Wu et al. (2007)). Then, we determine the inter-annual precipitation time variances ($\sigma_{r,s}^2$) from the detrended data. For each region $r$ we average means and variances across all data sets $s$ to obtain $\sigma_r^2$ and $\mu_r$. Finally, the standard deviation of natural variability in percent is obtained as $100 (\sigma_r/\mu_r)$. The last column in Table 1 expresses the derived regional natural variability.

---

[2]The time and resources needed for reaching a converged solution may exceedingly increase with the number of regions.

[3]The AOGCMs are BNU-ESM, CanESM, CSIRO-Mk3L-1-2, HadCM3, HadGEM2-ES, IPSL-CM5A-LR, MIROC-ESM, MPI-ESM-LR, and NorESM1-M.

## 2.3 Model

For our joint SRM-mitigation analysis, we utilize the integrated energy-economy-climate model MIND (Edenhofer et al. (2005)), which provides one of the simplest possible options to distinguish a renewable from a fossil sector and to include induced technological change. Compared to more advanced models that would distinguish electricity, a household, and a transport sector, it tends to underestimate mitigation costs by a factor of two (see, e.g., Edenhofer et al. (2014)). While its economy does not display any spatial resolution, it serves as one of the simplest possible models to project mitigation costs realistically. Hence, it can serve as a pedagogic model to mimic the essential economic-climatic aspects under investigation. We extend the model with respect to its climate diagnostics to include a spatially explicit resolution in terms of 'Giorgi regions' (Giorgi and Bi (2005)). This way, the SRM-side effect category 'SRM-induced regional climate mismatch' can be studied. In addition, we extend the model to include SRM as a control. We assume a reduction of the solar constant is a good approximation (Kalidindi et al. (2015)) of sulfur aerosol injection that is currently discussed as the most feasible SRM scheme. As the cost of SRM, we take the joint upper end of the expenses reported in The Royal Society's Report on Geoengineering the Climate (Shepherd (2009) and Klepper and Rickels (2011)): 0.02% gross world product as of 2010 per $W/m^2$. Even with this upper end, the costs of SRM are at least an order of magnitude smaller than the cost of mitigation (Edenhofer et al. (2014)).

Climate sensitivity is a crucial uncertain parameter, and some studies considered a log-normal probability density distribution for it (Lorenz et al. (2012); Neubersch et al. (2014); Roshan et al. (2019); Wigley and Raper (2001)). In this study, the model MIND is used in its deterministic setting with a climate sensitivity of 3°C. The time scale of the climate module has a distinct relationship with the climate sensitivity suggested by Lorenz et al. (2012). MIND employs the simplest climate module (one-box climate model) (Petschel-Held et al. (1999)). Nonetheless, Khabbazan and Held (2019) showed that a one-box climate model is a good emulator for fourteen tested AOGCMs (accurate to within 0.1°C for Representative Concentration Pathways, RCPs), provided the one-box climate model is tuned to the AOGCM's equilibrium climate sensitivity and transient climate response, and a certain time horizon (on the order of the time to peak radiative forcing) is not exceeded (see Khabbazan and Held (2019) for a detailed discussion). Therefore, according to Khabbazan and Held (2019), the results in this article can be interpreted as being influenced by a slightly larger climate response to forcing than intended. Hence, for the sake of completeness, in Sect. 4.3 we estimate the sensitivity of our results to the assumption made for climate sensitivity.[4]

## 2.4 Definition of scenarios

The analysis is based on the following scenarios (see Table 2): a) No-policy case (businessas-usual scenario ('BAU') where neither SRM nor mitigation is applied; hereby, in our context of the semi-conceptual MIND model, we do not distinguish BAU from baseline—in either case, no explicit climate policy is applied); b) 2°C target activated and SRM is not used ('TradCEA'); c) 2°C target is activated and SRM is not limited by regional contraints, hence complete ignorance of SRM side effects ('REF'); d) 2°C target plus all regional (precipitation) constraints are binding, and the extra admissible range ($EA$) is 5% of the standard

---

[4]Roshan et al. (2019) employed the MIND model developed here in its probabilistic version.

deviation, $\sigma_{r,\text{precip}}$ ('G0 5%'; 'G0' refers to 'Giorgi and Bi regions whereby zero regions are omitted as binding targets, in contrast to later modified applications'); e) Similar to c) but with 10% of the standard deviation ('G0 10%').

## 3 Results

Figure 3 displays the time evolution of normalized precipitation anomalies in the 26 regions four scenarios (BAU, REF, G0
5%, and G0 10%). We normalize the precipitation such that '1' is the constraint corresponding to the precipitation levels of a temperature anomaly of 2°C, and '0' indicates the constraint provided by the preindustrial precipitation levels. Note that in calculating the normalized precipitation guardrails for all scenarios, the extra admissible ranges are taken into account (see Sect. 2.1 for more details on the guardrails). We indicate these corridors as grey bands in Fig. 3. Figure 4 displays the effects on the global mean temperature by 'CO$_2$-forcing' (dotted lines), SRM-forcing (dashed lines), and the sum of both (solid lines)
for the four scenarios. Jointly, the two figures can be interpreted as follows.

In BAU, the 2°C target is transgressed (Fig. 4 (a)) and also the precipitation leaves the admissible corridors for most regions (Fig. 3(a)). Under unrestricted SRM usage in REF, the CO$_2$-contribution would mimic BAU, but SRM would avoid overshooting the 2°C target (Fig. 4 (b)). Due to its relatively low cost, SRM almost completely crowds out mitigation. For about half of the regions, however, precipitation transgresses the 2°C-compatible corridor (see Fig. 3 (b)). This demonstrates how the
definition of regional targets and subsequent G0 scenarios allows to preserve the value system encoded in the 2°C target when including SRM.

The precipitation corridors of all regions are activated in the G0 scenarios. By construction, for all of the regions, the precipitation trajectories stay confined to the grey band (see Fig. 3 (c) and (d)). Comparing G0 5% with G0 10%, one notices that the upper and lower bounds in G0 5% are touched 15 years earlier due to the smaller admissible range in G0 5% compared to G0
10%. SRM usage is highly restricted in the G0 5% and G0 10% scenarios compared to REF. As a result of this, the temperature anomaly peaks at 2°C and then declines, for example, to 1.5°C in G0 10%, which means SRM partly overcompensates the CO$_2$-effect (see Fig. 4 (c) and (d)) on global temperature.

This overcompensation can be explained from panels c) and d) in Fig. 3 in which the following four phases along the time axes can be identified. (i) No guardrail is active, and the paths mimic a BAU development (compared to panel a)). (ii) The 2°C
guardrail is active, SRM is utilized (as in the case depicted in Fig. 4 (b)), and the normalized precipitation anomaly declines for most regions. (iii) For one region, the upper ('1') normalized precipitation guardrail is reached, the normalized precipitation approaches the guardrails (either '0' or '1') for several regions, and the global mean temperature decreases simultaneously. (iv) The lower ('0') normalized precipitation guardrail is reached in a different region, and the system becomes quasi-stationary.

The first region reaching its upper normalized precipitation guardrails, and thereby starting phase 3, is 'AMZ' (Amazonia,
see Fig. 2). AMZ is characterized by a negative $c_{\text{CO}_2}$ and positive, but comparatively small $c_{\text{SRM}}$, which makes AMZ's absolute value of $R$ ($c_{\text{SRM}}/c_{\text{CO}_2}$=-0.14) one of the lowest among the regions. Precipitation in AMZ is continuing to increase in phase 2 as the increase due to further CO$_2$ emissions is merely compensated by SRM. Note that there are two regions whose $R$ is lower than AMZ; CNA ($R_{\text{CNA}}$=-0.11) and NAU ($R_{\text{NAU}}$=-0.07). Nevertheless, the comparably lower standard deviation of

natural variability (and hence, the extra admissible range, $EA$) in AMZ (4.42) than in CNA (8.93) and NAU (17.83) results in AMZ reaching its boundary of regional normalized precipitation faster than others.[5] . Once in AMZ the upper boundary of regional normalized precipitation levels is reached (start of phase 3), any $CO_2$-induced change in AMZ's precipitation needs to be compensated for by an SRM-induced contribution. Due to AMZ's small absolute $R$, more SRM forcing needs to be applied per unit of $CO_2$-induced radiative forcing than in phase 2 to stop AMZ's normalized precipitation trajectory from growing any further. Therefore, in phase 3, SRM overcompensates $CO_2$ in terms of its effects on the global mean temperature. Finally, one of the regions with larger absolute scaling coefficient ratio (in this case, 'SQF' (South Equatorial Africa)) hits the lower boundary of regional normalized precipitation, starting phase 4, where no further $CO_2$ emissions are admissible because because any attempt to compensate their temperature effect would result in a transgression of the precipitation corridor of at least one region.[6] Therefore, the 'G0' scenario is characterized by the interplay of scaling coefficients and precipitation standard deviations of two specific regions. Please note, however, that for different regional $R$s, there could be less than four phases, and the interplay could be between the global 2°C target itself and precipitation in one region. This would, e.g., be the case if the absolute $R$s of all regions are negative, as in our case, but all absolute values are larger than 1. If at least one of the regions has a positive $R$, an implementation of SRM could lead to a transgression of the region's precipitation guardrail even before the 2°C temperature limit is reached.

What are the economic effects of allowing for G0 (i.e., restricted SRM) instead of mitigation only? Figure 5 displays mitigation costs for a step-wise omission of regional precipitation guardrails. It is not a straightforward task to express policy-induced time-aggregated relative economic changes such as 'mitigation costs' in a consistent manner. We choose the option to utilize 'Balanced Growth Equivalent (BGE) values' (Anthoff and Tol (2009)). Policy-induced relative BGE changes represent relative changes of the initial and any future consumption of a stylized consumption path that are welfare-equivalent to the real consumption changes. The leftmost bars display the economic losses (disregarding impacts) induced by a 2°C policy without SRM usage (TradCEA). The economic losses for the G0 scenarios are shown by the following bars, continued by the scenarios when the guardrails of binding regions are disregarded one by one. The results indicate that 2/5 and 3/5 of the mitigation costs could be saved respectively in G0 5% and G0 10%. For someone who interprets mitigation costs as high, this could be an argument for employing SRM. For someone who perceives the scale of 1% of consumption loss as low, the mitigation cost would not be a reason to become interested in SRM.

---

[5]The absolute $R_r$ can also be perceived as the change in SRM-induced precipitation relative to $CO_2$-induced precipitation change for the same effect on global mean temperature. Because the extra admissible range is only activated when SRM comes into play, it can be adjusted according to the $R_r$ to measure the effective slimness of the regions' $EA$. For this measurement, we can define an effective admissible range ($EAR_r = R_r \cdot \sigma_{r,\mathrm{precip}}$). Therefore, AMZ would have the lowest $EAR$, and hence, it is the region that will most likely touch its precipitation guardrail when SRM is applied. One note is important to be mentioned: As there are chances that the deployment of SRM starts earlier than the temperature guardrail is touched, $EAR_r$ only indicates the likely candidates to touch their upper boundary of normalized precipitation faster than others.

[6]Depending on the signs of $c_{CO_2}$ and $EAR_r$, a larger absolute value of $EAR_r$ can also be perceived as how quickly region $r$ departs from its upper boundary of normalized precipitation. This is equivalent to say how quickly region $r$ may approach its lower boundary of normalized precipitation. Therefore, according to $EAR_{\mathrm{SQF}}$, SQF is likely to touch its lower boundary quicker than others if SRM is applied. However, as SRM deployment starts when the regions are relatively nearer to their upper boundary than their lower boundary, $R_r$ might be a better index than $EAR_r$ to signal which region may be quicker in reaching its lower boundary of normalized precipitation.

However, the picture can gradually change if society is willing to disregard the step-wise (economically) most binding corridor boundary and, hence, to 'sacrifice' the region that causes the strongest limitation of economic welfare gain. Progressively, more economic improvement could be gained (see Fig. 5, further right bars) when the region's guardrail that would deliver the largest economic welfare gain from one bar to the next is omitted. We choose the region to be omitted by asking which omission would cause the largest welfare gain. Figure 6 indicates the economic gain per region when the precipitation is allowed to leave the respective regional corridor. For G0 5% and G0 10%, the order of the first four regions whose guardrail should be omitted to gain the most is AMZ, SQF, CAM, and WAF. However, while these four regions are followed by CNA, MED, and CAS for G0 10%, in G0 5%, CAS, CNA, and SAH follow. Note that the precipitation changes in other regions are already within the guardrails in the REF scenario. Hence, there is no need to omit their guardrails to benefit welfare (see Fig. 3).

The order of regions may depend on the decision about the extra room for guardrails, as shown in panels a) and b). The reason is that with a tighter guardrail, while SRM is used less and mitigation must be employed more, the interplay of different scaling coefficients would likely cause different ordering. Nevertheless, if such an extra admissibility range is tighter, then the economic gain from its omission is higher. The same rule applies to the G0 scenarios, too. For example, the BGE loss in G0 5% is almost 60% higher than the BGE loss in G0 10%. Also, the BGE loss from omitting AMZ when the extra range is 5% of the natural variability is nearly double the cost for the same scenario when the extra room is 10% of the natural variability.

## 4 Sensitivity analysis

The results in the previous section were derived based on specific assumptions. Here we pick up some of the most critical assumptions and investigate alternative scenarios.

### 4.1 Extra room as a fraction of natural variability

Figure 7 depicts the BGE loss (%) for the G0 scenario when an addition of some fraction of the standard deviation of natural variability as the extra room varies from 0.05 (5%) to 0.5 (50%). The first two bars on the left, 0.05 and 0.1, are the same as the G0 scenarios in Fig. 5 (a) and (b), respectively. As can be expected, with larger admissibility ranges, the BGE loss decreases. When the extra range is 50% of the standard deviation of natural variability (the rightmost bar), the BGE loss is negligible (about -0.01% of BAU). However, such a decrease in BGE loss is not linear with respect to the increase in the extra range, but it is convex. For example, while the reduction in BGE loss is about 0.25% when the extra range changes from 0.05 to 0.1, the reduction in the BGE loss will decline to nearly 0.15% when the extra range changes from 0.1 to 0.15. Note that, according to the argument presented in Sect. 3, there is a chance that the order of regions changes.

### 4.2 Scaling coefficients

As discussed earlier, the SRM and $CO_2$ scaling coefficients determine the order of regions as well as when the guardrails in the G0 scenario are reached. However, the scaling coefficients depend on the AOGCM from which they are derived. The results in Sect. 3 were derived from the average value of scaling coefficients from nine AOGCMs. These data can also be used to derive

specific standard deviations for each scaling coefficient. Figure 8 shows the box plots for a Monte Carlo study on 1000 random, simultaneous variations in scaling coefficients and measures the BGE loss in the G0 scenario. In addition, the extra room in guardrails can increase in each scenario from 10% of the standard deviation of natural variability to 100% of it. In each G0 scenario, the sign of scaling coefficients and the binding regions in the four phases can be different.[7]

According to the Monte Carlo study, it is more likely that the random variation of the scaling coefficient results in a higher BGE loss in G0 scenarios. For example, while the BGE loss is about 0.45% in the deterministic results for an extra admissibility range equal to 10% of natural variability, the median of BGE loss in the Monte Carlo study can reach about 0.85%. In addition, with the higher extra ranges, the median of the BGE loss decreases. Nonetheless, similar to its deterministic case, the reduction in the median of the BGE loss is convex. Therefore, although for an extra admissibility range equal to 50% of natural variability

the BGE loss is negligible in the deterministic case, the BGE loss is about 0.4% in the Monte Carlo study. In other words, if the precipitation guardrails are considered, the likelihood of SRM use may still be low if no region's guardrail is about to be omitted. Note that the decision about SRM use may involve many more factors than just an economic study. However, our results at least call for attempts to better estimate the sensitivities of regional precipitation changes to SRM and global temperature increase.

## 4.3   Climate sensitivity

Figure 9 shows the BGE loss in TradCEA, G0 5%, and G0 10% scenarios with the climate sensitivity varying between 1.5°C to 5°C. As expected, with higher climate sensitivities, BGE losses for these scenarios increase rapidly. The 2°C target is not attainable without SRM when the climate sensitivity is equal and higher than 3.75°C for our model. Yet, by using SRM, the 2°C target is reachable with higher climate sensitivities. Nonetheless, the feasibility space depends on the extra room in the

guardrails. If the extra room is 10% of the natural variability, the 2°C target is still reachable when the climate sensitivity is below 5°C (not reachable at 5°C). However, if the extra room is only 5% of the natural variability, the 2°C target is only reachable when the climate sensitivity is below 4.25°C (not reachable at 4.25°C).

## 5   Conclusion

We performed a CEA (cost-effectiveness analysis) study where SRM (solar radiation management) and mitigation are simulta-

neously allowed as climate policy options. We investigated the minimal-cost mix of these options under certain environmental constraints (i.e., targets). As the key innovation, we defined a scheme to include one prominent side-effect category of SRM, regional climate pattern mismatches, in the integrated assessment, ethically consistent with the global mean temperature target. (By 'pattern mismatch' we refer to discrepancies in greenhouse gas- and SRM-induced spatial climate anomalies for the same global mean temperature change.) For this, we defined a metric to extend the functionality of global mean temperature targets

into a regime of SRM deployment. This extension is necessary as SRM destroys the relation of global mean temperature and regional climate known from greenhouse gas forcing. Hence, global mean temperature alone ceases to be a good proxy for the

---

[7]Here we do not go further into the discussion about the order of regions.

status of the climate system. Accordingly, we augment the global mean temperature target by equivalent regional temperature targets under pure greenhouse gas forcing. Thereby, the analysis does not rely on the global mean temperature target alone, and it can directly employ the equivalent regional targets when SRM is considered. We suggest somewhat arbitrary but ethically consistent regional targets by asking what climate those regions would have experienced in a 2°C warmer world without any

SRM. Accordingly, for economic optimization, we would allow only for those scenarios which, for any region, would stay in the same interval as generated by global mean temperature anomalies between 0 and 2°C. From all possible SRM-induced climate mismatches, we chose precipitation as a particularly significant one.

Without accounting for SRM side effects, it would crowd out mitigation due to its comparatively low costs for achieving the 2°C target (compared to the original mitigation costs). However, when only one single regional climate variable (in our case

'precipitation') is required to stay within regional bounds compatible with the global 2°C target, the cooling contribution by SRM needs to be reduced to a value lower than 0.4°C (0.8°C) depending on the allowed overshoot (5% (10%) of the standard deviation of annual mean regional precipitation). However, 2/5 or 3/5 of mitigation costs could be saved due to the steepness of the mitigation cost curve. A more significant role for SRM would be possible if the guardrails of a few regions were relaxed. The ordering of regions presented in this article might provide a way to support the trade-off between the relaxations of global

versus regional targets. Nonetheless, the order of regions whose omission brings about the most economic welfare gain depends on the magnitude of $CO_2$- and SRM-induced effects on precipitation as well as the normative decision on the upper and lower bounds of the precipitation guardrails.

We would see SRM and mitigation as complements if only a global climate policy were implemented within decades. The results showed that in our model, the 2°C target is not attainable without SRM when the climate sensitivity is equal or higher

than 3.75°C. From the mitigation policy perspective, this is equivalent to saying that abatement is delayed until, even for more centered values of climate sensitivity such as 3°C, the emission budget becomes exhausted. Then we would necessarily need some climate engineering to comply with the 2°C target (Lawrence et al. (2018)). If the potential for carbon dioxide removal were exhausted, some amount of SRM would become indispensable when the 2°C target still should be reached. Nonetheless, the feasibility space of SRM depends on the exact definition of the guardrails. The tighter they are, the earlier SRM becomes

useless to comply with the targets.

We need to point to a series of caveats of the analysis. The assumptions made in constructing regional scaling patterns for precipitation may oversimplify the complex hydrological effects of greenhouse gases and SRM. Hence, we need to emphasize that regional precipitation guardrails can only be interpreted as necessary, not as sufficient conditions for decisions about the use of SRM. Yet, our Monte Carlo study shows that the random variation of scaling coefficients would likely result in larger

economic losses when temperature and precipitation targets are binding. Furthermore, annual mean precipitation is only one possible driver of regional climate impacts in addition to temperature, evaporation, or intra-annual changes. Finally, our model does not include carbon dioxide removal options yet. Hence, the above-mentioned economic gains through SRM must be interpreted as upper limits of cost savings.

Here, we demonstrate that someone who pushes for SRM to reach the 2°C target should carefully consider this target's

consequences when in part achieved by SRM. A regionally explicated climate variable (such as precipitation) reduces the

usage of SRM to 1/3 even if one allows for a transgression of regional targets by 10% of the standard deviation of natural variability. Inclusion of further side effects of SRM would result in additional reduction factors.

## DATA

The output of nine AOGCMs was supplied by the scientists from GCESS, Beijing Normal University (John Moore's Group).
Three data sets of annual Giorgi regions precipitation were supplied by M. Büchner and K. Frieler from the Potsdam Institute of Climate Impact Research (PIK). All climate model outputs used in this paper are available online on The Earth System Grid Federation (ESGF) webpages.

## Authors' contributions

M.M.K. and M.S. performed the numerical analyses, wrote most of the code, and integrated SRM into MIND, M.M.K. optimized the GAMS code, wrote the code for ordering regions, designed and conducted the sensitivity analysis, and prepared most of the visualization, M.S. and E.R derived spatially resolved SRM-coefficients, E.R. calculated natural variability of precipitation, H.S. provided the system-analytic link between SRM experiments and integrated assessment, H.H. and E.R. explained the non-monotonic temperature response of Fig. 4, H.H. triggered this work and supplied the concept of extending temperature targets to SRM climate risks, H.H. and H.S. developed the 4 phases model, M.M.K., H.S., and H.H. developed the explanation of the regions' selection mechanism, H.H. and M.M.K. wrote most of this report.

## Competing interests

The authors declare that they have no conflicts of interest.

## Acknowledgments

E.R. has been supported by the DFG grant HE 555812-1 within the DFG priority program 'Climate engineering – Risks, Challenges, Opportunities? (SPP1689). H.S. acknowledges support under the DFG grant SCHM 2158/4-1. We are thankful to scientists from GCESS, Beijing Normal University (John Moore's Group), for supplying the output nine of AOGCMs. Also, we are thankful to M. Büchner and K. Frieler from the Potsdam Institute of Climate Impact Research (PIK) for supplying data sets of annual Giorgi regions precipitation.

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

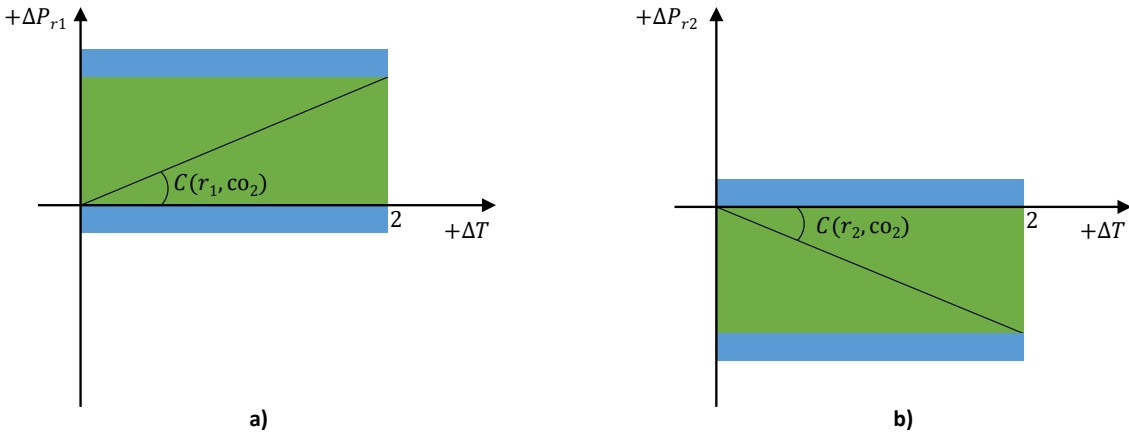

**Figure 1. Schematic of precipitation guardrails for two hypothetical regions** $r1$ **and** $r2$. $r1$ is characterized by a positive $CO_2$ (Greenhouse-gas-driven) scaling coefficient ($C(r1, CO_2) > 0$), and $r2$ is characterized by a negative $CO_2$ scaling coefficient ($C(r2, CO_2) < 0$). The graphs show regional precipitation vs. global mean temperature, the latter in °C. The green bands in panel a) and panel b) define the regular admissible area for precipitation change. The extra admissible areas ($EA$), as a fraction of regional standard deviation of natural variability, are demonstrated in blue.

| | Giorgi region | $c_{CO_2}$[%/K] | $\sigma_{c_{CO_2}}$[%/K] | $c_{SRM}$[%/K] | $\sigma_{c_{SRM}}$[%/K] | $R_r = c_{SRM}/c_{CO_2}$ | $\sigma_{r,precip}$[%] |
|---|---|---|---|---|---|---|---|
| ALA | Alaska | 5.51 | 1.12 | -6.38 | 1.09 | -1.16 | 4.84 |
| AMZ | Amazonia | -1.35 | 2.39 | 0.18 | 2.43 | -0.14 | 4.42 |
| CAM | Central America | -4.11 | 1.87 | 2.54 | 1.32 | -0.62 | 7.19 |
| CNA | Central North-America | -0.37 | 3.27 | 0.04 | 3.18 | -0.11 | 8.75 |
| CAS | Central Asia | 1.02 | 2.02 | -2.31 | 1.52 | -2.25 | 8.93 |
| CSA | Central South-America | 1.01 | 0.84 | -1.84 | 1.01 | -1.83 | 8.66 |
| EAF | East Africa | 4.78 | 2.86 | -5.71 | 3.17 | -1.20 | 6.45 |
| EAS | East Asia | 1.81 | 1.34 | -2.36 | 1.26 | -1.30 | 5.65 |
| ENA | East North-America | 1.10 | 1.10 | -1.75 | 1.13 | -1.59 | 5.85 |
| EQF | Equatorial Africa | 4.52 | 3.49 | -6.15 | 4.07 | -1.36 | 11.36 |
| GRL | Greenland | 4.66 | 1.08 | -5.37 | 1.00 | -1.15 | 5.50 |
| MED | Mediterranean | -4.01 | 1.34 | 3.54 | 1.18 | -0.88 | 7.28 |
| NAS | North Asia | 5.26 | 1.14 | -6.06 | 1.10 | -1.15 | 3.71 |
| NAU | North Australia | -0.31 | 3.47 | 0.02 | 3.37 | -0.07 | 17.83 |
| NEE | North-East Europe | 3.08 | 1.27 | -4.68 | 1.44 | -1.52 | 6.57 |
| NEU | Northern Europe | 2.19 | 0.77 | -3.47 | 1.01 | -1.59 | 6.14 |
| SAF | South Africa | -1.78 | 1.20 | 1.74 | 1.12 | -0.98 | 15.79 |
| SAH | Sahara | 2.99 | 10.07 | -2.15 | 10.71 | -0.72 | 21.80 |
| SAS | South Asia | 1.66 | 1.28 | -2.43 | 1.25 | -1.46 | 5.47 |
| SAU | South Australia | -2.16 | 1.44 | 1.76 | 1.72 | -0.81 | 15.19 |
| SEA | South-East Asia | 1.74 | 1.60 | -2.46 | 1.55 | -1.41 | 8.43 |
| SQF | South Equatorial Africa | 0.04 | 1.63 | -0.91 | 1.78 | -22.28 | 5.74 |
| SSA | South South-America | 0.93 | 0.75 | -1.63 | 0.77 | -1.74 | 7.84 |
| TIB | Tibetan Plateau | 4.13 | 1.56 | -5.17 | 1.86 | -1.25 | 14.62 |
| WAF | West Africa | 0.11 | 1.39 | -0.57 | 1.17 | -4.99 | 6.29 |
| WNA | West North-America | 1.93 | 2.96 | -2.41 | 2.77 | -1.25 | 11.64 |

**Table 1. Scaling characteristics of Giorgi regions.**

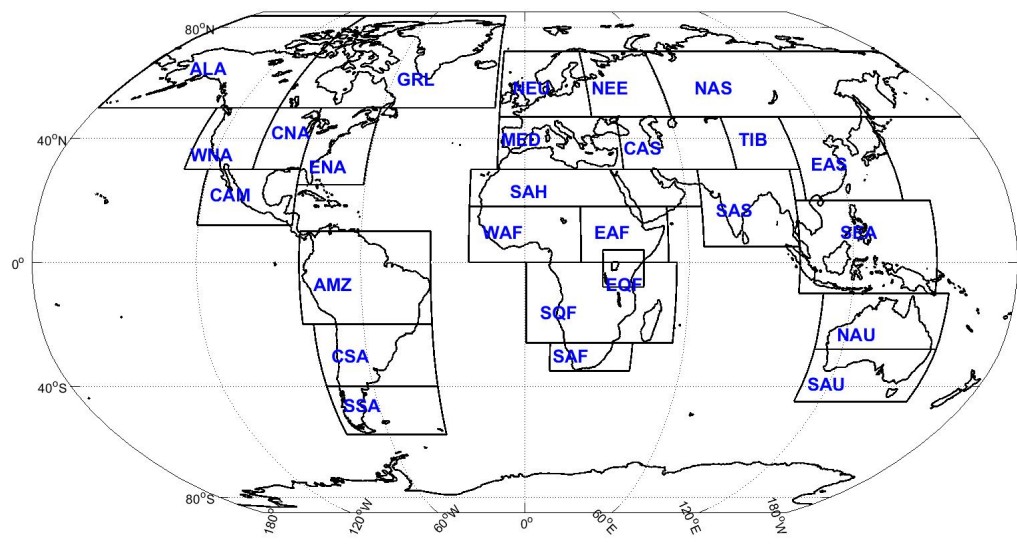

**Figure 2. Spatial resolution of our analysis.** 'Giorgi regions' Giorgi and Bi (2005).

| Scenario | SRM | Mitigation | 2°C Target | Precipitation Guardrails | $AE = 0.05 \cdot \sigma_{r,\text{precip}}$ | $AE = 0.10 \cdot \sigma_{r,\text{precip}}$ |
|---|---|---|---|---|---|---|
| BAU | | | | | ✓ | |
| TradCEA | | ✓ | ✓ | | ✓ | |
| REF | ✓ | ✓ | ✓ | | ✓ | |
| G0 5% | ✓ | ✓ | ✓ | ✓ | ✓ | |
| G0 10% | ✓ | ✓ | ✓ | ✓ | | ✓ |

**Table 2. Scenarios and their characteristics.**

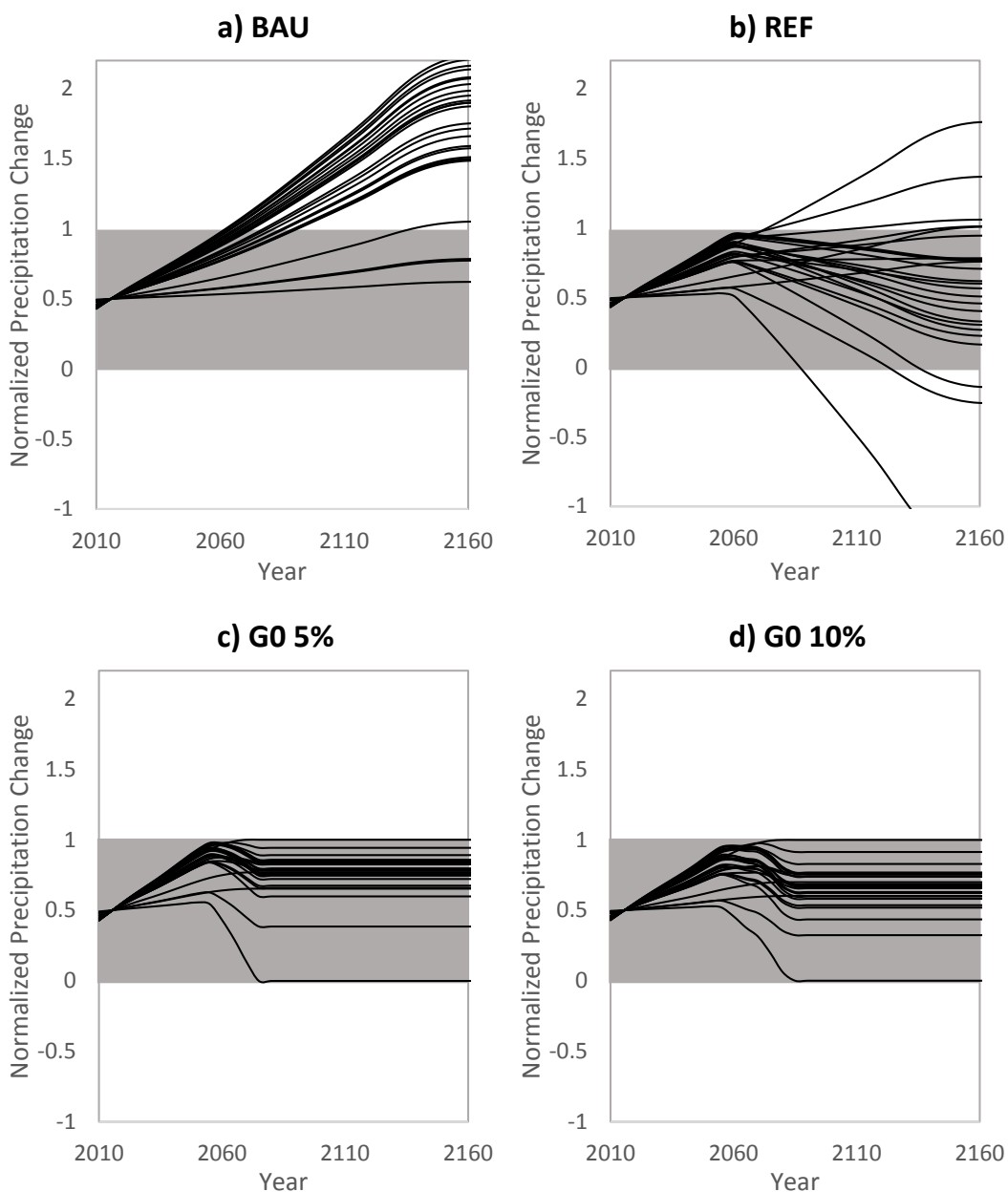

**Figure 3. Normalized precipitation change.** a): No policy ('BAU'); b): 2°C target activated and unlimited usage of SRM ('REF'); c) all Giorgi regions' precipitation guardrails being activated when the extra admissible area is 5% of the standard deviation ('G0 5%'); d) similar to c) but with 10% of the standard deviation ('G0 10%'). For the 26 Giorgi regions and different policy scenarios, precipitation change is normalized such that '1' is the constraint which corresponds to the precipitation levels of a temperature anomaly of 2°C, and '0' equals the constraint which is determined by the preindustrial precipitation levels. Note that in calculating the normalized precipitation guardrails for G0 5% and G0 10% scenarios, the extra admissible areas are taken into account (see Sect. 2.1 for more details on the guardrails). These corridors are indicated as grey bands.

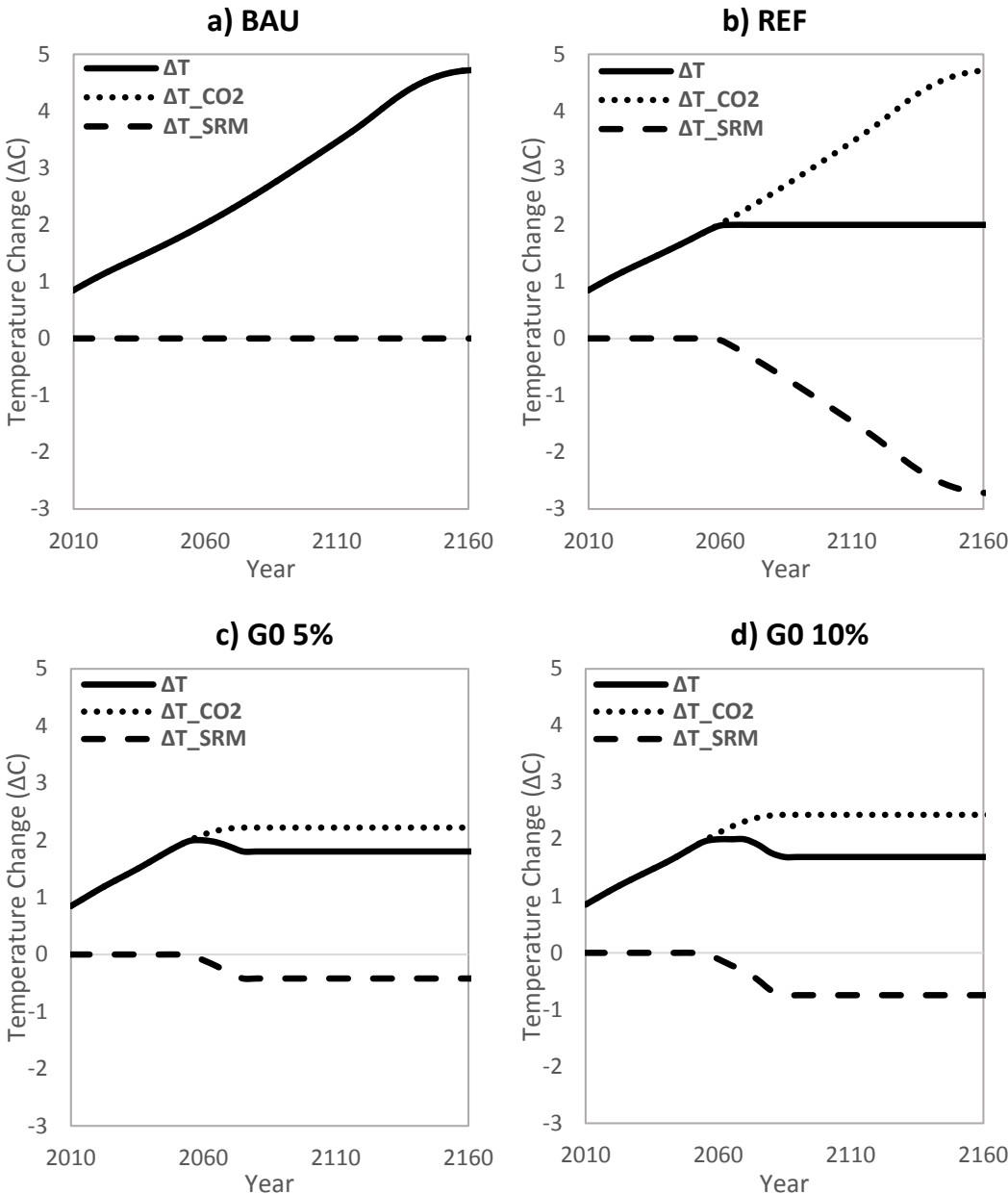

**Figure 4. Global mean temperature response to SRM and carbon dioxide forcing.** Dotted lines are the resulting effects on global mean temperature for temperature response to $CO_2$-forcing; The dashed lines are the resulting effects on global mean temperature for temperature response to SRM-forcing; the solid lines are the sum of both dotted and dashed lines. a): No policy ('BAU'); b): 2°C target activated and unlimited usage of SRM ('REF'); c) all Giorgi regions' precipitation guardrails being activated when the extra admissible area is 5% of the standard deviation ('G0 5%'); d) similar to c) but with 10% of the standard deviation ('G0 10%').

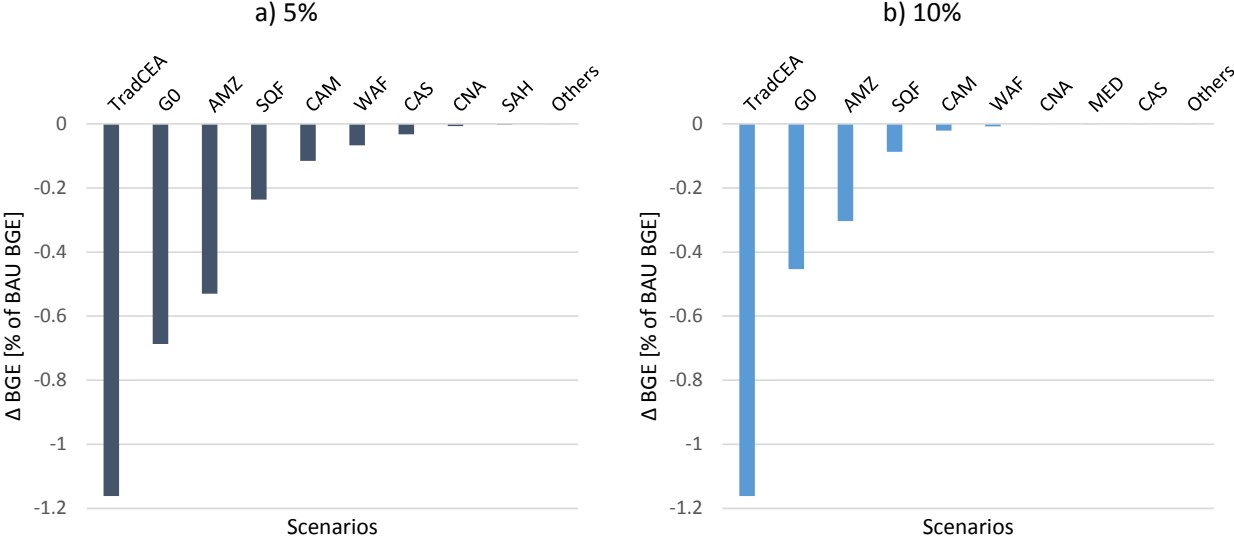

**Figure 5. Mitigation costs for a step-wise omission of regional precipitation guardrails.** 'TradCEA' is a 2°C policy without SRM usage; 'G0' is a 2°C policy when SRM is included, and all regional precipitation guardrails are active; 5% and 10% denote the fraction of the standard deviation of natural variability as the extra admissible area. BGE values attributed to a region are the loss in comparison to a no policy (BAU – business as usual) scenario if the analysis is henceforth not constrained by the precipitation guardrails of the respective region. The acronyms of regions are defined in Table 1.

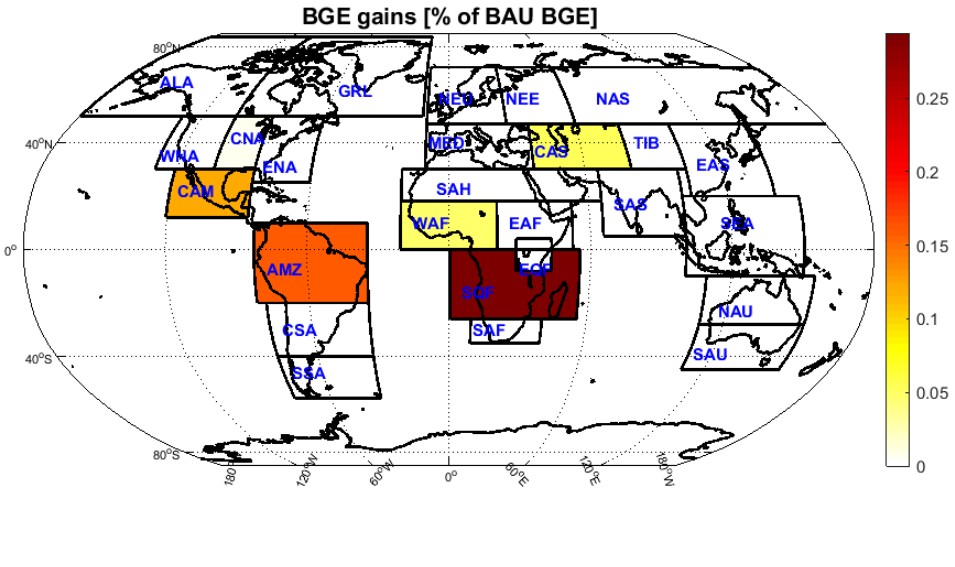

(a) G0 5%

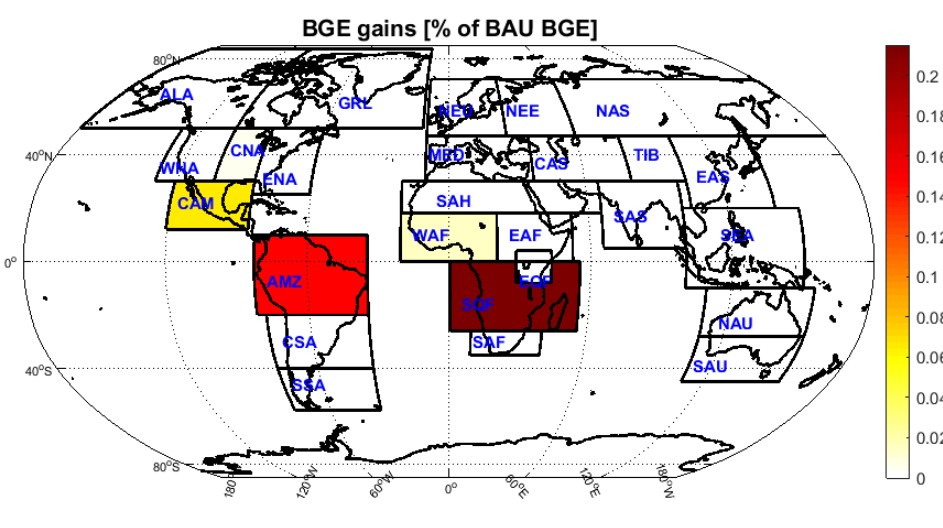

(b) G0 10%

**Figure 6. Economic gains from omission on regional guardrails.** Darker colors indicate more economic gain per region when the precipitation leaves the respective regional corridor. 5% and 10% denote the fraction of the standard deviation of natural variability as the extra admissible area.

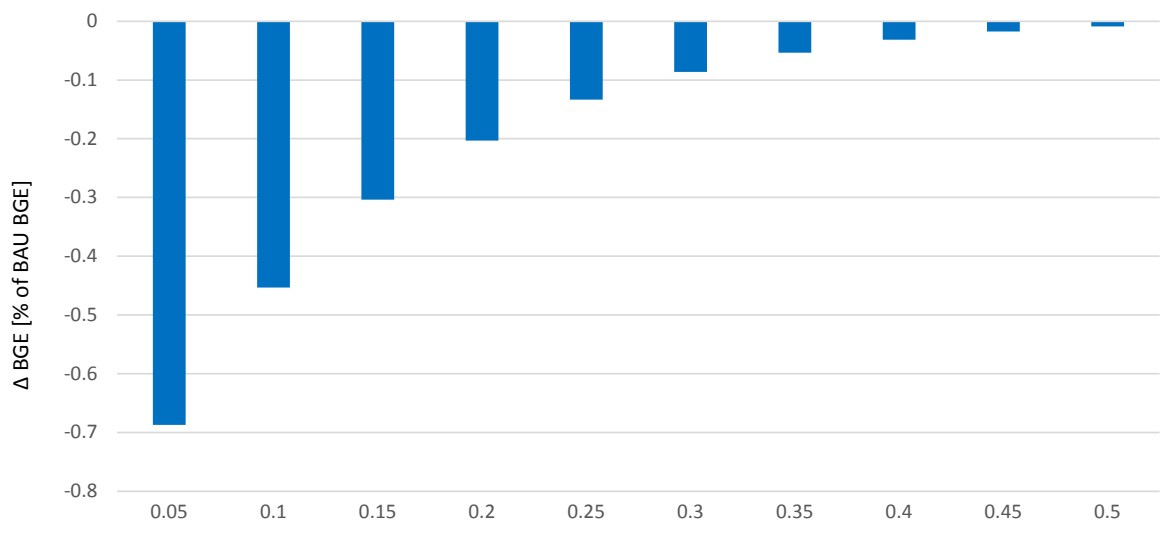

**Figure 7. Sensitivity analysis on the fraction of standard deviation of natural variability added to the admitted precipitation corridor.** The extra admissible is increased from 5% of the standard deviation of natural variability to 50% of the standard deviation of natural variability. The leftmost two bars (0.05 and 0.1) correspond to the G0 5% and G0 10% in Figure 5.

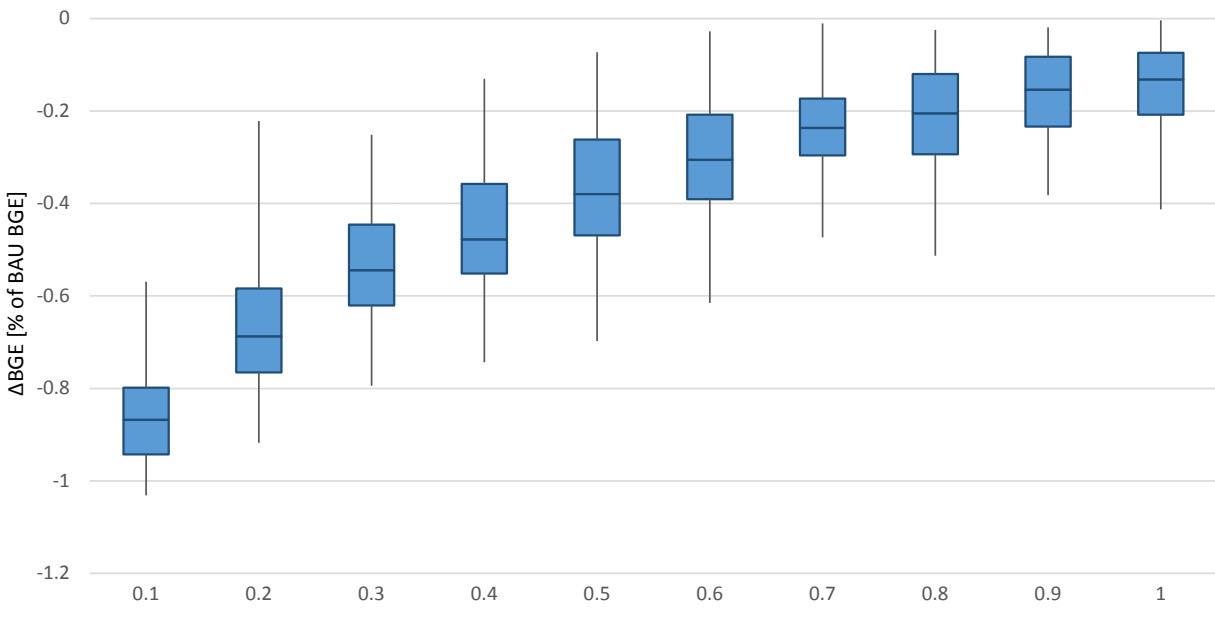

Fraction of standard deviation of natural variability added to the admitted precipitation corridor

**Figure 8. Monte Carlo study of SRM and CO₂ scaling coefficients.** The box and whisker plots show the minimum, first quartile, median, third quartile, and maximum BGE losses. The boxes are drawn from the first quartiles to the third quartiles. The horizontal lines go through the boxes at the medians. The whiskers go from each quartile to the minimums or maximums.

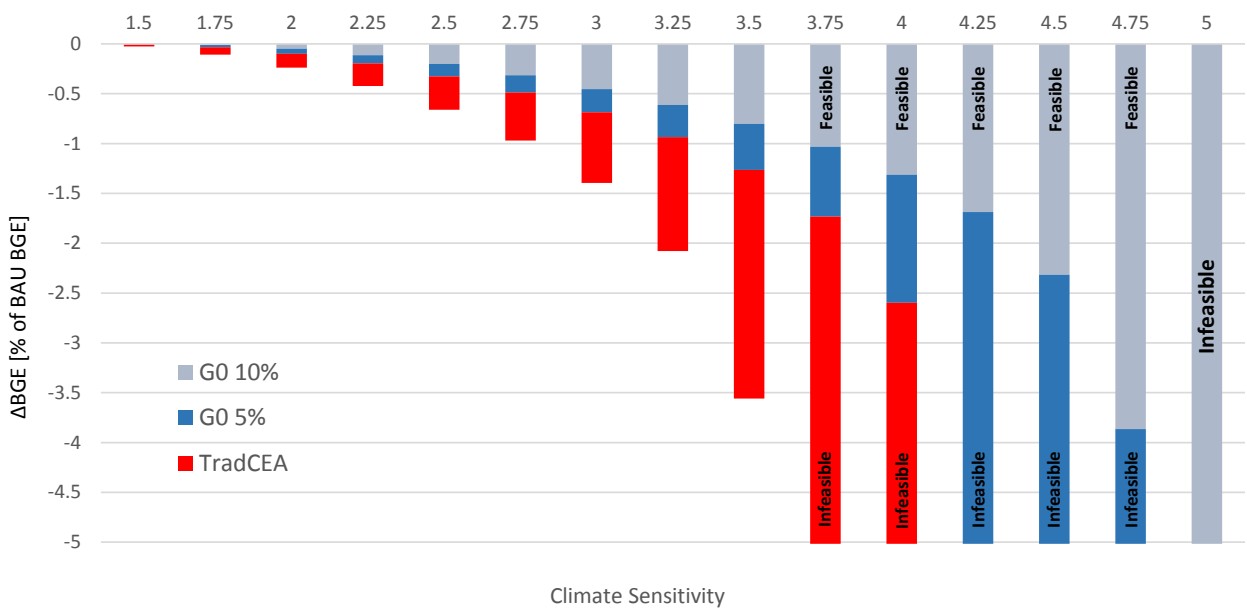

**Figure 9. Sensitivity analysis on climate sensitivity.** The grey bars show the BGE losses for the G0 10% scenarios. The blue bars show the BGE losses for the G0 5% scenarios. The red bars show the BGE losses for the TradCEA scenarios. For all scenarios, the climate sensitivity ranges from 1.5°C to 5°C.

**Appendix**

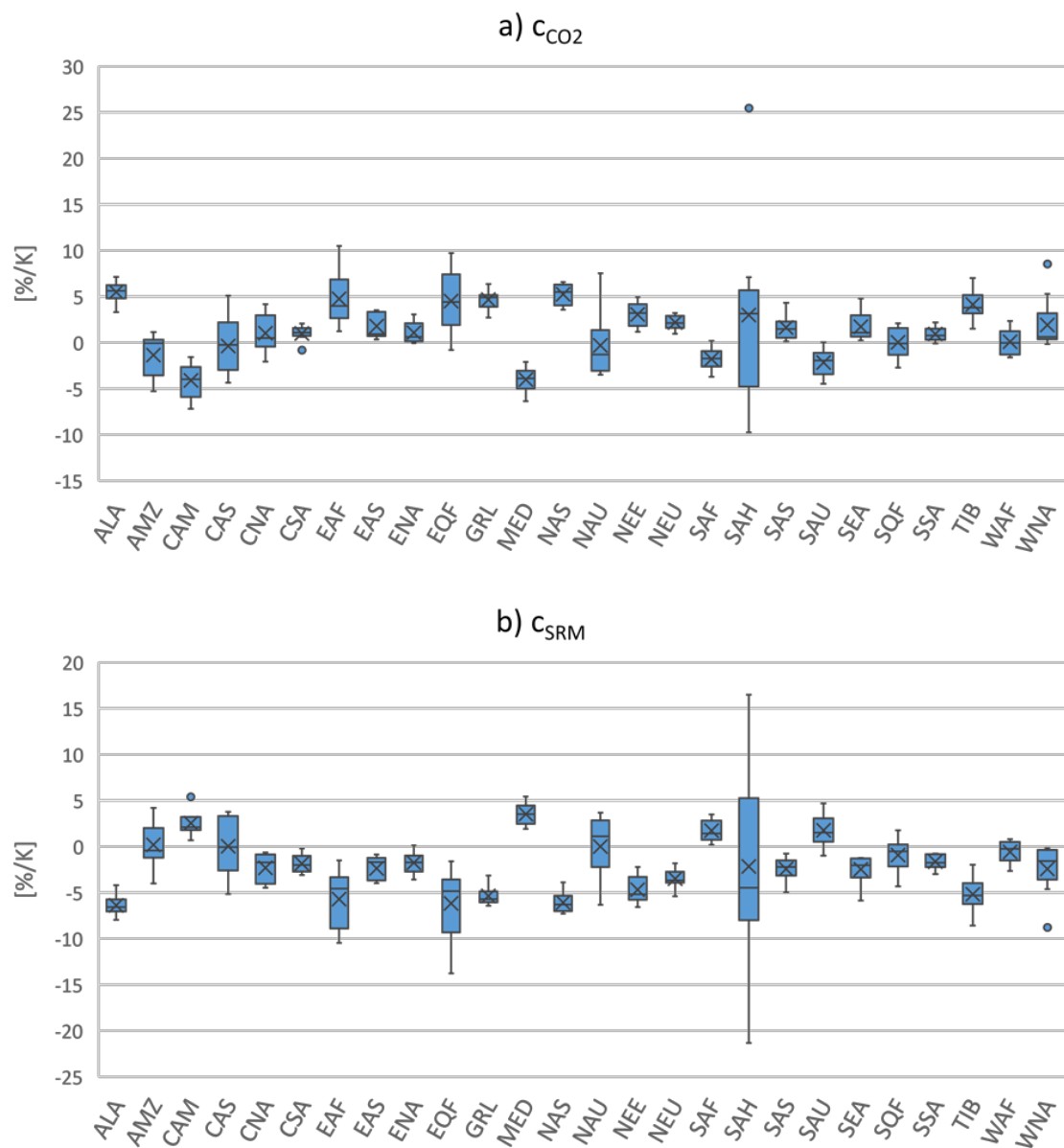

**Figure A1. Variation of scaling coefficients for regions from nine AOGCMs.** The boxes are drawn from the first quartiles to the third quartiles. The horizontal lines go through the boxes at the medians. The crosses show the averages. The whiskers go from each quartile to the minimums or maximums. The dots represent the outliers.