# Peer review of "How to Combine Solar Geoengineering and Mitigation under Climate Targets?"

_Earth System Dynamics, 2020_

## Referee Comment (RC1) · Anonymous Referee #1 · 1 Feb 2021

A review of the manuscript "Trade-offs of Solar Geoengineering and Mitigation under ClimateTargets" by Khabbazan et al.

The manuscript considers a joint strategy of emission reductions and SRM, where the use of SRM is constrained by regional precipitation 'guardrails'. These guardrails are hypothetical or illustrative, determined by scaling the regional mean precipitation observed in a 2C scenario without SRM. The authors then use an energy-economy-climate model MIND to calculate scenarios of such joint strategies.

General comments:

1. The overall concept of the paper is good. It provides some new (according to my knowledge) and concrete proposals for thinking about the negative impacts that SRM

might have. Although the guardrails used in the scenarios are merely hypothetical, they illustrate the concept nicely.

2. On one hand, I really like that the climate impact is represented not only as the mean temperature change (which is certainly a proxy for many different things). But on the other, adding only precipitation as a more specific measure of climate change impact might be problematic. The authors should discuss this issue further.

3. My main concern is that the concept and modeling approach need to be presented much more clearly. When I started to read the results, I was unsure what the different cases were and how they were calculated. Section 2 focuses on describing verbosely why things are done in the chosen way, i.e. the underlying rationale; but is too vague on describing what is actually done. I provide some points in the specific comments, but the authors should aim more generally for a clear exposition of their approach.

4. The methods section describes the used methods (to some extent, see the comment above), but not the calculated cases. The definitions should also be covered under section 2 (rather than in the results, section 3). For example, how was the level and time profile of SRM determined? Was it uniform across the globe or was it determined somehow regionally? Did it start only in 2060, as Figure 4 suggests? These things are not stated anywhere, and it's hard to interpret the meaning of your results without such information.

5. Another main concern is that the results need to be presented in a better way. Now they are quite uninformative, and as a result, the paper misses it's potential. This requires substantial reworking of the figures and text. Please see the specific comments for some ideas.

6. Just a thought, but what might be interesting to see is the correspondence between the 'mitigation costs' (loss in economic growth) and some measure of additional precipitation that some chosen SRM or guardrail level entails. Another thing would be to understand which regions experience the SRM precipitation effects, and to which di-

rection these effects are pointing (Figure 3 didn't provide this information, although it could have).

7. The manuscript would benefit from condensing the text, which is quite verbose at times. This would improve the clarity and readability of the text. A language check would also be a useful thing to do. I would also suggest using a colleague to check whether the paper is understandable for an outsider.

Specific comments:

P2, L5: You mention that 'any meaningful assessment must include a risk-risk trade-off'. However, there's no uncertainty or risk considered in your analysis, if I'm correct. So, does this mean that your analysis is not meaningful?

P2, L15: The statement on this paper being the first to use "an integrated analysis of SRM and mitigation in-line with the '2°C temperature targe" is inaccurate, as such analyses have been presented already earlier (e.g. Smith and Rasch, Climatic Change 2013; Ekholm and Korhonen, Climatic Change 2016; Emmerling and Tavoni ,Env. Res. Econ, 2018). There is also a review "Evaluating climate geoengineering proposals in the context of the Paris Agreement temperature goals" by Lawrence et al. (2018) that is related to this discussion. That is, the authors need to define this paper's novelty more carefully (which is discussed more in the next paragraph).

P3, L3: What does "admissible" mean here, as it can be interpreted in very different? The citation (Bruckner et al. 2008) is not very helpful here without a further explanation. Is there a better citation than a conference paper?

P3, L6: Somewhat unclear what is meant by "necessary condition" and "keeping the 2°C target in order" (retain?).

P3, L7: The question "How much regional precipitation change, as an example of a climatic change other than temperature, would someone, who has already accepted up to 2°C of global warming, accept?" is interesting, but too unclear to get what you

mean by it. Who is this 'someone', what means 'to accept 2°C' or 'to accept an amount of precipitation change'?

P3, L9: The next sentence ("If we were able to confine regional climate change...") needs also some clarification. Maybe break it into two or three sentences and explain the idea piece-by-piece? P3, L16-26: Please state more clearly and carefully whether you mean the average or the standard deviation of precipitation change in each context.

P3, L17: Why it's called a 'scaling coefficient'? I don't think this is used to scaling anything, but it represents the response in (average?) precipitation due to an increase in global mean temperature. Why it's a function with the second argument as 'CO2'? Should it rather be Delta T? (And, these maybe could be better expressed as indices, not function arguments.)

P3, L25: Please state more explicitly, that the extra admissible area is not based on any calculation of the impacts that the additional change in precipitation could bring about. That is, these are used only in an illustrative manner (which is perfectly ok, as long as it's stated clearly).

P3, L28: Introduce briefly what the Giorgi regions stand for or represent.

P4, L15-17: Here's a good example of a sentence that is too verbose and unclear. Please clarify.

P4, L24: Is "parabolic fit of the time evolution" a second-order polynomial trend?

P4, L29 onwards: Again, there is too much 'meta' talk about the model, but it's not clear what the model actually covers and models. What it means that the model "co-shaped the mitigation chapter of the Stern Report", and does it matter here? A comparison to "more advanced models" is not very useful, if I don't know how advanced or simple the MIND model is.

P5, L5: You extended the model, but how exactly? What regions it had earlier? A new regional split in this kind of a model is not a simple thing to do, so it requires some

further explanation.

P5, L12: Climate sensitivity seems not to be your focus in this work, so a shorter statement of this and the used value will do fine. Also, the defense for the simple climate module in MIND is perhaps a bit too long. Ok, it's simple and therefore inaccurate, but you perform sensitivity analysis to check this.

P5, L24-27: It is very hard to understand what the cases mean and how they have been calculated, as these weren't explained in the methods section. What does '2°C target activated' mean, or 'all regional constraints are binding'? How have the mitigation and SRM contributions been determined? This all is unclear for me.

P5, L30: So, the normalized values of 0 and 1 are not the max and min from the 'observations' between preindustrial and 2C scenario, but including the 5% or 10% leeway? Then, in the lower subfigures of Figure 3, the normalized levels 0 and 1 imply different absolute levels, am I correct? Please clarify.

P6, L10 onwards: This description is hard to follow. The panels c) and d) in figure 3 are too similar, and the measure (normalized precipitation change) too abstract to get much insights from it (please see also my comment regarding Fig. 3). I merely see (from Fig. 4) that the looser guardrail allows for a stronger level of SRM.

P7, L13: This is in one way a good idea, but quite off the point regarding the paper's topic. Additionally, the considered values seem rather arbitrary.

P9, L17-18: Wasn't the idea of guardrails exactly to manage risks of SRM, i.e. that the change in precipitation due to SRM will be small compared to that which occurs due to climate change itself. In this regard, it would be good to compare more explicitly the trade-off between mitigation cost and recipitation change (guardrail level).

P9, L25: I belief this is more of a 'feature' of your model, rather than a 'hard, cold truth'. I guess there's a plenty of scenarios that remain below 2C even with a higher climate sensitivity. You could perhaps just point out to the literature where SRM is discussed

from a 'last resort' perspective, if emission reductions are postponed too much, there's tipping points or such.

P10, L6-8: I don't think the final statements reflect your results. (Plus, the first sentence is rather vague.) I think SRM had a notable role in lowering the mitigation costs in your cases, whereas the precipitation change due to SRM was still confined into a quite moderate interval.

Figure 1: The figure would benefit from a few improvements, so that the main idea would come more clearly through to the reader. Please add degree Celsius symbol for the number two. Perhaps include some indication that the blue is some chosen fraction of the green (if I understood correctly). Is the y-axis the average of precipitation change? If so, this could be indicated there.

Figure 3: I don't find the figure (when presented in this way) particularly informative. All it says that with BAU, climate change continues strong, with no restrictions on SRM the precipitation guardrails are breached in some regions (but not which), and with the guardrails in place – well, precipitation stays within the guardrails. The normalized results are too abstract to have much meaning for an outsider, while there is now information on which line is which region (I assumed they represent different regions, but this wasn't said expressly in the caption).

Figure 5: This figure is quite confusing. There are too many acronyms. The caption says 'mitigation costs', but the y axis is '% of BAU BGE'. After some thinking I figured the connection between these, and how SRM with guardrails (case G0) lowers the mitigation costs (reduces the decrease in economic growth). But then, are the first two columns global figures; and which case the regional columns correspond to? Do only a handful of regions experience mitigation costs? Is mitigation virtaully free in e.g. North America and Europe? This confuses me a lot. (Additionally: why TradCEA was not listed as one of the calculated cases?)

Figures 3, 4 and 5: Maybe include more informative titles for the subfigures than the

'codes' (BAU, REF etc.)?

---

## Referee Comment (RC2) · Anonymous Referee #2 · 24 Feb 2021

Review on the manuscript entitled "Trade-offs of Solar Geoengineering and Mitigation under Climate Targets" by Mohammad M. Khabbazan, Marius Stankoweit, Elnaz Roshan, Hauke Schmidt, and Hermann Held submitted to Earth System Dynamics

The authors have discussed a temperature target-based approach for mitigating the impact of global warming on future climate using solar radiation management. The authors have introduced a concept for an integrated Solar Radiation Management (SRM) analysis and mitigation in-line with the 2°C temperature target. Therefore the paper is novel but limited by the approach adopted for the study. Authors have considered temperature and precipitation simultaneously in their approach for mitigation of the impact of CO2 rise. However, it is not specified explicitly in the manuscript why temperature and precipitation have been selected for the analysis. However, after reading cited

references, for example, Edenhofer et al., 2005; Gorgi and Bi (2005), readers may infer about it. Therefore I feel the paper needs to be rewritten for better clarity and readability.

I think the manuscript needs a major revision before its publication.

My specific comments are mentioned below.

1. Some of the earlier studies (Bala et al. 2008 and so on thereafter) indicated that alteration in solar forcing might offset temperature changes or hydrological changes from greenhouse warming but could not cancel both at once. However, the authors have designed an integrated analysis of SRM for mitigation of global warming impact in-line with the '2°C temperature target by constraining the regional precipitation changes. In line with this, the authors should provide the appropriate justification for designing these experiments to make the manuscript's physical basis more robust.

2. The manuscript's title indicates the manuscript will discuss the trade-off between solar geoengineering and mitigation under climate targets. However, the tread-off/optimization aspect has not been discussed clearly, even though it can be inferred from the discussions presented.

3. Page 3 Lines 7-12 "Here we ask: 'How much regional precipitation change, as an example of a climatic change other than temperature, would someone, who has already accepted up to 2°C of global warming, accept?' If we were able to confine regional climate change to the intervals of climate variables that would be spanned by ramping the global mean temperature anomaly (as against its pre-industrial value) up from zero to 2°C, we could augment the 2°C target by this exact set of intervals as the more fundamental target." needs simplification and clarity.

4. Page 4 lines 3-5: "For the scaling coefficients, we diagnose annual mean regional precipitation changes from linear pattern scaling (Ricke et al. (2010)) which are driven as a linear superposition (Ban-Weiss and Caldeira (2010)) of greenhouse-gas-induced

and SRM-induced changes in global mean temperature." The authors need to provide a detailed explanation of the procedure followed for obtaining scaling coefficients.

5. The authors have used the outputs of nine atmosphere-ocean general circulation models (AOGCMs). However, details have not been produced in the manuscript. These models' configurations (physics, horizontal and vertical resolution, etc.) are different. It is not clear that how authors have normalized the impact of configuration changes? If they have considered regional averages only (over Giorgi regions), CSRM and CCo2 should be measured in [%/K/km2]. It is not clear how the regional average will represent the regional change. The methodology adopted in this manuscript is quite confusing or needs a better explanation. A small change in a large area will have reasonably more impact than a large change in relatively smaller areas. Will conclusions presented in this study will differ if authors consider the regional averages by normalizing areal coverage of the regions specified in the manuscript.

6. Page 5 Lines 24-27: "Figure 3 shows normalized precipitation change for the 26 Giorgi regions for a): no-policy case (business as usual scenario 25 ('BAU') where neither SRM nor mitigation is applied); b): 2°C target activated and unlimited admissible SRM level ('REF'); c): precipitation changes when all regional constraints are binding and the extra admissible area is 5% of the standard deviation ('G0 5%'); d) similar to c) but with 10% of standard deviation ('G0 10%')". I feel a realistic analysis and experimental design are required.

7. The results presented in the manuscript rewritten with clarity and should be robust because many hypothetical statements have been mentioned in the result section. Further, the authors indicated that they had not considered the measures related to the reduction of CO2 due to the impact of other policies and absorption by oceans. Therefore the results from this paper should be considered as the upper limit. However, it is not discussed in the abstract. Therefore, for the readers who will read only the abstract section, this manuscript's conclusions will be misleading.

8. Authors should provide the links of datasets used for the present research work and comply with the journal's data policy.

---

## Author Comment (AC1) · 30 Mar 2021

**Note:** For the sake of transparency and a point-by-point reply, each original comment by Anonymous Referee #1 is repeated in italic shape, and the Author Response follows it in upright shape.

*A review of the manuscript "Trade-offs of Solar Geoengineering and Mitigation under Climate Targets" by Khabbazan et al.*

*The manuscript considers a joint strategy of emission reductions and SRM, where the use of SRM is constrained by regional precipitation 'guardrails'. These guardrails*

*are hypothetical or illustrative, determined by scaling the regional mean precipitation observed in a 2C scenario without SRM. The authors then use an energy-economyclimate model MIND to calculate scenarios of such joint strategies.*

*General comments:*

*1. The overall concept of the paper is good. It provides some new (according to my knowledge) and concrete proposals for thinking about the negative impacts that SRM might have. Although the guardrails used in the scenarios are merely hypothetical, they illustrate the concept nicely.*

We would like to thank the reviewer for this overall assessment. And yes, we agree that our guardrails we introduced are merely hypothetical. However, our claim is that they are ethically unique in the sense that they can readily be derived from the global $2°$C target. They provide the straightforward regionalization of that global temperature target. We strive at making this point clearer in a new version of our ms.

*2. On one hand, I really like that the climate impact is represented not only as the mean temperature change (which is certainly a proxy for many different things). But on the other, adding only precipitation as a more specific measure of climate change impact might be problematic. The authors should discuss this issue further.*

Our argument consists of two steps. (i) Due to the climate pattern mismatch between SRM and CO2-induced climate changes, we need to generalize a global target to a set of regional targets. (ii) What variables to choose for regional climate targets? Temperature (Asseng et al. (2011)) and precipitation (Portmann et al. (2010)) are highly relevant for agricultural productivity. Hereby the pattern mismatch of precipitation is of a larger order of magnitude than that of temperature (Kravitz et al. (2014)). We are not claiming that temperature and precipitation are the only relevant climate predictors for agricultural productivity or the functionality of ecosystems in general, but we acknowledge precipitation limits as a necessary boundary condition within a

target-based framework.

*3. My main concern is that the concept and modeling approach need to be presented much more clearly. When I started to read the results, I was unsure what the different cases were and how they were calculated. Section 2 focuses on describing verbosely why things are done in the chosen way, i.e. the underlying rationale; but is too vague on describing what is actually done. I provide some points in the specific comments, but the authors should aim more generally for a clear exposition of their approach.*

Thank you very much for this comments. To address your valid comment here, we will clarify the description by adding equations and probably a table to show the differences between cases.

*4. The methods section describes the used methods (to some extent, see the comment above), but not the calculated cases. The definitions should also be covered under section 2 (rather than in the results, section 3). For example, how was the level and time profile of SRM determined? Was it uniform across the globe or was it determined somehow regionally? Did it start only in 2060, as Figure 4 suggests? These things are not stated anywhere, and it's hard to interpret the meaning of your results without such information.*

We are very grateful to you for these detailed comments and questions. We will definitely clarify all of these points (including level and time profile of SRM, geographical application of SRM, and starting points) in the revised version. We will also transfer the definitions from Section 3 to Section 2.

*5. Another main concern is that the results need to be presented in a better way. Now they are quite uninformative, and as a result, the paper misses it's potential. This requires substantial reworking of the figures and text. Please see the specific comments*

*for some ideas.*

Thank you very much for this comments. We will use the opportunity of providing a revised version for presenting our results in a better way as suggested.

*6. Just a thought, but what might be interesting to see is the correspondence between the 'mitigation costs' (loss in economic growth) and some measure of additional precipitation that some chosen SRM or guardrail level entails. Another thing would be to understand which regions experience the SRM precipitation effects, and to which direction these effects are pointing (Figure 3 didn't provide this information, although it could have).*

Thanks for this nice comment. Please note that because the economy module in our model is global, i.e., there are no regional economic specifications, therefore we are unable to provide further information on regions. We hoped that Figure 7, specifically, could show the trade offs between the "global mitigation cost" and "additional precipitation." Nevertheless, we will use your comments and explain your point more clearly in the main text and elaborate on how a multi-regional model results could be different from ours.

*7. The manuscript would benefit from condensing the text, which is quite verbose at times. This would improve the clarity and readability of the text. A language check would also be a useful thing to do. I would also suggest using a colleague to check whether the paper is understandable for an outsider.*

We are grateful to you for this comments. We will try to use equations and a table for clarifying things and avoid verbose texts wherever possible. Please note that, before submission, we had asked a colleague to read our paper. However, we will further ask colleagues to assure the readability of our manuscript. Also, we will definitely ask a native English editor for further language check.

*Specific comments:*

*P2, L5: You mention that 'any meaningful assessment must include a risk-risk tradeoff'. However, there's no uncertainty or risk considered in your analysis, if I'm correct. So, does this mean that your analysis is not meaningful?*

Here we utilized a certain lingo to be found in the SRM community. From the reviewer's reaction we diagnose that this can cause misunderstandings. We refer to the inclusion of side-effects of SRM most of which are uncertain, some even deeply uncertain. Hereby we utilize 'uncertain' in the sense of IPCC AR5 WGIII Annex I: It includes any situation of lack of knowledge, in particular it also covers probabilistic knowledge (IPCC (2014)).

*P2, L15: The statement on this paper being the first to use "an integrated analysis of SRM and mitigation in-line with the '2C temperature targe" is inaccurate, as such analyses have been presented already earlier (e.g. Smith and Rasch, Climatic Change 2013; Ekholm and Korhonen, Climatic Change 2016; Emmerling and Tavoni ,Env. Res. Econ, 2018). There is also a review "Evaluating climate geoengineering proposals in the context of the Paris Agreement temperature goals" by Lawrence et al. (2018) that is related to this discussion. That is, the authors need to define this paper's novelty more carefully (which is discussed more in the next paragraph).*

We are grateful for this hint and will certainly cite this and further CEA-based literature on CEA, including the one cited in our twin-paper on SRM Roshan et al. (2019).. Based on that discussion we will make much clearer what the innovative aspect of our ms is. In fact, none of the above articles tackles a joint mitigation / SRM integrated assessment which includes regional pattern mismatches induced by SRM. Smith and Rasch (2013) focus on the main effect of SRM and answer how much SRM is to be added for a given greenhouse gas emission scenario if a certain global mean

temperature target is to be complied with. Ekholm and Korhonen (2016) consider the effect of the timing uncertainty about when SRM can be deployed. Emmerling and Tavoni (2018) investigate the effects of uncertainty about SRM effectiveness and anticipated future learning about it on first-period decisions. Lawrence et al. (2018) present a meta-assessment on whether SRM can play a significant role for compliance with temperature targets at all. None of them discusses a new metric as ours to extend the ethics represented by the 2°C target to the assessment of one category of side-effect of SRM.

*P3, L3: What does "admissible" mean here, as it can be interpreted in very different? The citation (Bruckner et al. 2008) is not very helpful here without a further explanation. Is there a better citation than a conference paper?*

'Admissible' means 'in compliance with a certain environmental target or further constraints'. We'll cite Kriegler and Bruckner (2004) instead.

*P3, L6: Somewhat unclear what is meant by "necessary condition" and "keeping the 2C target in order" (retain?).*

We want to express that we add regional environmental constraints, without removing the global 2°C target. We strive at expressing this clearer in a new ms.

*P3, L7: The question "How much regional precipitation change, as an example of a climatic change other than temperature, would someone, who has already accepted up to 2C of global warming, accept?" is interesting, but too unclear to get what you mean by it. Who is this 'someone', what means 'to accept 2C' or 'to accept an amount of precipitation change'?*

Apparently, this statement is currently too condensed. It represents the key conceptual

innovation of the ms. By this ms, we present a solution to the conceptual evaluation problem that SRM destroys the correlation between the global indicator 'global mean temperature' and regional climate. This means, also for those decision-makers who accepted compliance with a global mean temperature target as a guideline for decision-making, this can no longer work as a guideline, once SRM is applied. By this article, hence, we strive at closing this evaluation gap. We ask: 'What are the regional analogues, if not to say 'derivatives' of the global mean temperature interval [0...T*] which would have been acceptable for a proponent of a T*-target? Which regional climate change does correspond to T*?' (With in our case, T*=2°C.)

*P3, L9: The next sentence ("If we were able to confine regional climate change...") needs also some clarification. Maybe break it into two or three sentences and explain the idea piece-by-piece?*

Thanks for this hint. We hope that this sentence is now clearer based on our responses above. Nevertheless, we will clarify this as suggested in more sentences.

*P3, L16-26: Please state more clearly and carefully whether you mean the average or the standard deviation of precipitation change in each context.*

This adds to the above point. Apparently, the key innovation must be explained clearer. We will do so also with adding the relevant equations. And we will explain clearer what is the role of the standard deviation of precipitation.

*P3, L17: Why it's called a 'scaling coefficient'? I don't think this is used to scaling anything, but it represents the response in (average?) precipitation due to an increase in global mean temperature. Why it's a function with the second argument as 'CO2'? Should it rather be Delta T? (And, these maybe could be better expressed as indices, not function arguments.)*

Here 'scaling' refers to 'pattern scaling' – see e.g. Frieler et al. (2012) and Osborn et al. (2016). The idea is that regional climate can in first order Taylor expansion be predicted from a few global variables such as $CO_2$-induced temperature change vs SRM-induced temperature change. Again, we think, adding the relevant equations will help.

*P3, L25: Please state more explicitly, that the extra admissible area is not based on any calculation of the impacts that the additional change in precipitation could bring about. That is, these are used only in an illustrative manner (which is perfectly ok, as long as it's stated clearly).*

We will do so.

*P3, L28: Introduce briefly what the Giorgi regions stand for or represent.*

Surely, we will do so.

*P4, L15-17: Here's a good example of a sentence that is too verbose and unclear. Please clarify.*

Thanks for the comment. We will do so.

*P4, L24: Is "parabolic fit of the time evolution" a second-order polynomial trend?*

Yes. Thanks for this note. We will correct it in the revised version.

*P4, L29 onwards: Again, there is too much 'meta' talk about the model, but it's not clear what the model actually covers and models. What it means that the model "co-shaped the mitigation chapter of the Stern Report", and does it matter here? A comparison to*

*"more advanced models" is not very useful, if I don't know how advanced or simple the MIND model is.*

We will expand on this. Our main point is that we describe in what sense it is a more realistic model than Nordhaus' DICE model, and how it compares to more complex climate-energy-economy models.

*P5, L5: You extended the model, but how exactly? What regions it had earlier? A new regional split in this kind of a model is not a simple thing to do, so it requires some further explanation.*

Thanks for this comment. Please note that the economic module in our model is not regionalized, and only the climate module includes regional disparities. Nevertheless, we will explain this point more precisely in the method section with appropriate equations. Please note that this "splitting approach" is possible, because from the economic module we only infer the mitigation costs. As we will explain clearer, here the globally aggregated approach is of sufficient complexity to substitute for spatially resolved economic models.

*P5, L12: Climate sensitivity seems not to be your focus in this work, so a shorter state-ment of this and the used value will do fine. Also, the defense for the simple climate module in MIND is perhaps a bit too long. Ok, it's simple and therefore inaccurate, but you perform sensitivity analysis to check this.*

Thank you very much for your comment. However, we prefer to keep it in, as such a paragraph was requested by other reviewers in the past for several times.

*P5, L24-27: It is very hard to understand what the cases mean and how they have been calculated, as these weren't explained in the methods section. What does '2C target*

*activated' mean, or 'all regional constraints are binding'? How have the mitigation and SRM contributions been determined? This all is unclear for me.*

Thank you very much for pointing out to this issue which apparently needs more explanation in our manuscript. We intend to use a table in method section for further clarification.

*P5, L30: So, the normalized values of 0 and 1 are not the max and min from the 'observations' between preindustrial and 2C scenario, but including the 5% or 10% leeway? Then, in the lower subfigures of Figure 3, the normalized levels 0 and 1 imply different absolute levels, am I correct? Please clarify.*

Yes, you are perfectly correct. We will clarify this in the revised manuscript.

*P6, L10 onwards: This description is hard to follow. The panels c) and d) in figure 3 are too similar, and the measure (normalized precipitation change) too abstract to get much insights from it (please see also my comment regarding Fig. 3). I merely see (from Fig. 4) that the looser guardrail allows for a stronger level of SRM.*

We will try to make Figure 3 and 4 more informative, by adding vertical grids to the figure and more explanation of the effects.

*P7, L13: This is in one way a good idea, but quite off the point regarding the paper's topic. Additionally, the considered values seem rather arbitrary.*

Thank you very much for your comment. Please note that from our point of view this issue is one of the key messages on the paper showing the tradeoffs between mitigation and SRM which can reflect itself as the tradeoffs between temperature target and the precipitation target as well due to our unique method described in this paper. We will definitely make this point clearer.

*P9, L17-18: Wasn't the idea of guardrails exactly to manage risks of SRM, i.e. that the change in precipitation due to SRM will be small compared to that which occurs due to climate change itself. In this regard, it would be good to compare more explicitly the trade-off between mitigation cost and recipitation change (guardrail level).*

We are thankful to you for this comments. We definitely assume that our explanation of the economic module was not clear enough. Yet, although we assume that we have addressed the tradeoff between mitigation cost and SRM, specifically, in Figure 7, this comment has appeared in this review frequently, we still leave a chance for misunderstanding from our side. Therefore, we will be eternally grateful to the reviewer and the editorial board if we are granted an extension of the open discussion for receiving more comments from the reviewer on our response, and especially, on this issue.

*P9, L25: I belief this is more of a 'feature' of your model, rather than a 'hard, cold truth'. I guess there's a plenty of scenarios that remain below 2C even with a higher climate sensitivity. You could perhaps just point out to the literature where SRM is discussed C5 from a 'last resort' perspective, if emission reductions are postponed too much, there's tipping points or such.*

We will do so.

*P10, L6-8: I don't think the final statements reflect your results. (Plus, the first sentence is rather vague.) I think SRM had a notable role in lowering the mitigation costs in your cases, whereas the precipitation change due to SRM was still confined into a quite moderate interval.*

Noted – we will condition our statement more precisely on the regional targets'

tolerance levels.

*Figure 1: The figure would benefit from a few improvements, so that the main idea would come more clearly through to the reader. Please add degree Celsius symbol for the number two. Perhaps include some indication that the blue is some chosen fraction of the green (if I understood correctly). Is the y-axis the average of precipitation change? If so, this could be indicated there.*

Thanks for the detailed comments. We will do so.

*Figure 3: I don't find the figure (when presented in this way) particularly informative. All it says that with BAU, climate change continues strong, with no restrictions on SRM the precipitation guardrails are breached in some regions (but not which), and with the guardrails in place – well, precipitation stays within the guardrails. The normalized results are too abstract to have much meaning for an outsider, while there is now information on which line is which region (I assumed they represent different regions, but this wasn't said expressly in the caption).*

We would like to keep it in. However, we will explain it more carefully. It describes at a glance the effects of qualitatively different scaling coefficients (or ratios thereof) on regional scenarios and their interaction with various targets.

*Figure 5: This figure is quite confusing. There are too many acronyms. The caption says 'mitigation costs', but the y axis is '% of BAU BGE'. After some thinking I figured the connection between these, and how SRM with guardrails (case G0) lowers the mitigation costs (reduces the decrease in economic growth). But then, are the first two columns global figures; and which case the regional columns correspond to? Do only a handful of regions experience mitigation costs? Is mitigation virtaully free in e.g. North America and Europe? This confuses me a lot. (Additionally: why TradCEA was not*

*listed as one of the calculated cases?)*

Thanks a lot for the comment. We hope that by our explanations above some of the concerns, especially on the regional mitigation cost, are already addressed. Nevertheless, we will strive at making these points clearer in the revised version.

*Figures 3, 4 and 5: Maybe include more informative titles for the subfigures than the 'codes' (BAU, REF etc.)?*

This is a nice and valid suggestion, thank you. We will do so.

**References**

Asseng, S., Foster, I., and Turner, N. C.: The impact of temperature variability on wheat yields, Global Change Biology, 17, 997–1012, 2011.

Ekholm, T. and Korhonen, H.: Climate change mitigation strategy under an uncertain Solar Radiation Management possibility, Climatic Change, 139, 503–515, 2016.

Emmerling, J. and Tavoni, M.: Climate engineering and abatement: A 'flat'relationship under uncertainty, Environmental and resource economics, 69, 395–415, 2018.

Frieler, K., Meinshausen, M., Mengel, M., Braun, N., and Hare, W.: A scaling approach to probabilistic assessment of regional climate change, Journal of Climate, 25, 3117–3144, 2012.

IPCC. Climate change 2014: Mitigation of climate change. Working group III contribution to the fifth assessment report of the intergovernmental panel on climate change, 2014.

Kravitz, B., MacMartin, D. G., Robock, A., Rasch, P. J., Ricke, K. L., Cole, J. N., Curry,

C. L., Irvine, P. J., Ji, D., Keith, D. W., et al.: A multi-model assessment of regional climate disparities caused by solar geoengineering, Environmental Research Letters, 9, 074 013, 2014.

Kriegler, E. and Bruckner, T.: Sensitivity analysis of emissions corridors for the 21st century, Climatic change, 66, 345–387, 2004.

Lawrence, M. G., Schäfer, S., Muri, H., Scott, V., Oschlies, A., Vaughan, N. E., Boucher, O., Schmidt, H., Haywood, J., and Scheffran, J.: Evaluating climate geoengineering proposals in the context of the Paris Agreement temperature goals, Nature communications, 9, 1–19, 2018.

Osborn, T. J., Wallace, C. J., Harris, I. C., and Melvin, T. M.: Pattern scaling using ClimGen: monthly-resolution future climate scenarios including changes in the variability of precipitation, Climatic Change, 134, 353–369, 2016.

Portmann, F. T., Siebert, S., and Döll, P.: MIRCA2000—Global monthly irrigated and rainfed crop areas around the year 2000: A new high-resolution data set for agricultural and hydrological modeling, Global biogeochemical cycles, 24, 2010.

Roshan, E., Khabbazan, M. M., and Held, H.: Cost-Risk Trade-Off of Mitigation and Solar Geoengineering: Considering Regional Disparities Under Probabilistic Climate Sensitivity, Environmental and Resource Economics, pp. 1–17, 2019.

Smith, S. J. and Rasch, P. J.: The long-term policy context for solar radiation management, Climatic Change, 121, 487–497, 2013.

---

## Author Comment (AC2) · 30 Mar 2021

**Note:** For the sake of transparency and a point-by-point reply, each original comment by Anonymous Referee #2 is repeated in italic shape, and the Author Response follows it in upright shape.

*Review on the manuscript entitled "Trade-offs of Solar Geoengineering and Mitigation under Climate Targets" by Mohammad M. Khabbazan, Marius Stankoweit, Elnaz Roshan, Hauke Schmidt, and Hermann Held submitted to Earth System Dynamics*

We are very grateful to the reviewer for providing such a detailed comments on our
manuscript, which will certainly add to the quality of our paper.

*The authors have discussed a temperature target-based approach for mitigating the impact of global warming on future climate using solar radiation management. The authors have introduced a concept for an integrated Solar Radiation Management (SRM) analysis and mitigation in-line with the 2C temperature target. Therefore the paper is novel but limited by the approach adopted for the study. Authors have considered temperature and precipitation simultaneously in their approach for mitigation of the impact of CO2 rise. However, it is not specified explicitly in the manuscript why temperature and precipitation have been selected for the analysis.*

Thank you very much for this comment. We will strive at a clearer description of this point. Temperature (Asseng et al. (2011)) and precipitation (Portmann et al. (2010)) are highly relevant for agricultural productivity.

*However, after reading cited references, for example, Edenhofer et al., 2005; Gorgi and Bi (2005), readers may infer about it. Therefore I feel the paper needs to be rewritten for better clarity and readability.*

*I think the manuscript needs a major revision before its publication.*

*My specific comments are mentioned below.*

*1. Some of the earlier studies (Bala et al. 2008 and so on thereafter) indicated that alteration in solar forcing might offset temperature changes or hydrological changes from greenhouse warming but could not cancel both at once. However, the authors have designed an integrated analysis of SRM for mitigation of global warming impact inline with the '2C temperature target by constraining the regional precipitation changes. In line with this, the authors should provide the appropriate justification for designing these experiments to make the manuscript's physical basis more robust.*

This is an excellent point. We will add the mathematical framework in an explicit form. The key point of our ms is to extend the decision-analytic framework of global mean temperature-based decision-making to a situation were global mean temperature ceases being a good predictor of regional climate.

*2. The manuscript's title indicates the manuscript will discuss the trade-off between solar geoengineering and mitigation under climate targets. However, the treadoff/ optimization aspect has not been discussed clearly, even though it can be inferred from the discussions presented.*

Thanks for your comment. We realize that the title in fact does not properly reflect our main point any more. We will change it accordingly – see also the above point.

*3. Page 3 Lines 7-12 "Here we ask: 'How much regional precipitation change, as an example of a climatic change other than temperature, would someone, who has already accepted up to 2C of global warming, accept?' If we were able to confine regional climate change to the intervals of climate variables that would be spanned by ramping the global mean temperature anomaly (as against its pre-industrial value) up from zero to 2C, we could augment the 2C target by this exact set of intervals as the more fundamental target." needs simplification and clarity.*

Noted. We expect that adding the mathematical framework will deliver the necessary clarity. The above §represents the key innovation of our paper and apparently requires reformulation. It is about the extension of a temperature target to a situation in which it ceases being a good predictor for regional climate.

*4. Page 4 lines 3-5: "For the scaling coefficients, we diagnose annual mean regional precipitation changes from linear pattern scaling (Ricke et al. (2010)) which are driven as a linear superposition (Ban-Weiss and Caldeira (2010)) of greenhouse-gas-induced*
*and SRM-induced changes in global mean temperature." The authors need to provide a detailed explanation of the procedure followed for obtaining scaling coefficients.*

Surely, this will be added.

*5. The authors have used the outputs of nine atmosphere-ocean general circulation models (AOGCMs). However, details have not been produced in the manuscript. These models' configurations (physics, horizontal and vertical resolution, etc.) are different. It is not clear that how authors have normalized the impact of configuration changes? If they have considered regional averages only (over Giorgi regions), CSRM and CCo2 should be measured in [%/K/km2]. It is not clear how the regional average will represent the regional change. The methodology adopted in this manuscript is quite confusing or needs a better explanation. A small change in a large area will have reasonably more impact than a large change in relatively smaller areas. Will conclusions presented in this study will differ if authors consider the regional averages by normalizing areal coverage of the regions specified in the manuscript.*

Thank you for this nice comment. We will present our data as well as AOGCMs in a properer and detailed way. Regarding the methodology to calculate CSRM and CCo2, we will explain precisely the procedure.

*6. Page 5 Lines 24-27: "Figure 3 shows normalized precipitation change for the 26 Giorgi regions for a): no-policy case (business as usual scenario 25 ('BAU') where neither SRM nor mitigation is applied); b): 2C target activated and unlimited admissible SRM level ('REF'); c): precipitation changes when all regional constraints are binding and the extra admissible area is 5% of the standard deviation ('G0 5%'); d) similar to c) but with 10% of standard deviation ('G0 10%')". I feel a realistic analysis and experimental design are required.*

As our analysis is in the tradition of explicating the consequences of ethical assump-

tions (see e.g., IPCC AR5 WGIII Ch6 (IPCC (2014))) and not of predictions, we are unsure what 'realistic analysis' does refer to. We would be grateful if the discussion phase could be utilized for a clarification on this particular issue or in general on our entire response.

*7. The results presented in the manuscript rewritten with clarity and should be robust because many hypothetical statements have been mentioned in the result section. Further, the authors indicated that they had not considered the measures related to the reduction of CO2 due to the impact of other policies and absorption by oceans. Therefore the results from this paper should be considered as the upper limit. However, it is not discussed in the abstract. Therefore, for the readers who will read only the abstract section, this manuscript's conclusions will be misleading.*

We will sharpen the abstract accordingly. We indeed want to study the inclusion of one important category of side-effect of SRM into an ethical framework coherent with a temperature target. This is our innovation and its effects can be studied best if applied to the combination SRM+mitigation.

*8. Authors should provide the links of datasets used for the present research work and comply with the journal's data policy.*

We will comply with journal's data policy and present a proper reference to our data.

**References**

Asseng, S., Foster, I., and Turner, N. C.: The impact of temperature variability on wheat yields, Global Change Biology, 17, 997–1012, 2011.

IPCC. Climate change 2014: Mitigation of climate change. Working group III contribution to the fifth assessment report of the intergovernmental panel on climate change, 2014.

Portmann, F. T., Siebert, S., and Döll, P.: MIRCA2000—Global monthly irrigated and rainfed crop areas around the year 2000: A new high-resolution data set for agricultural and hydrological modeling, Global biogeochemical cycles, 24, 2010.

---

## Author Comment (AC3) · 15 Jun 2021

We are very grateful to the reviewers for the concise and constructive reports on our MS *"Trade-offs of Solar Geoengineering and Mitigation under Climate Targets."* We have made a point-by-point reply to the reviewers' comments, which can be found under https://doi.org/10.5194/esd-2020-95-AC1 and https://doi.org/10.5194/esd-2020-95-AC2. Here, we present our Final Response, which includes our main plan for revising our MS upon permission. Our Final Response here can be seen as a summary of our detailed responses to the reviewers' comments.

1. We strive at making the point clearer that while our guardrails are merely hypothetical, they are ethically unique in the sense that they can readily be derived

from the global 2°C target. Hence, they provide the straightforward regionalization of that global temperature target.

2. Based on the following two arguments: (i) Due to the climate pattern mismatch between SRM and CO2-induced climate changes, we need to generalize a global target to a set of regional targets, and (ii) Temperature (Asseng et al. (2011)) and precipitation (Portmann et al. (2010)) are highly relevant for agricultural productivity, the pattern mismatch of precipitation is of a larger order of magnitude than that of temperature (Kravitz et al. (2014)). Therefore, we will clarify that temperature and precipitation are the only relevant climate predictors for agricultural productivity or the functionality of ecosystems in general, but we acknowledge precipitation limits as a necessary boundary condition within a target-based framework.

3. We will clarify Section 2 by adding equations and probably a table to show the differences between cases. The key point of our ms is to extend the decision-analytic framework of global mean temperature-based decision-making to a situation where global mean temperature ceases being a good predictor of regional climate.

4. We will clarify all of the points about including the level and time profile of SRM, geographical application of SRM, and starting points in the revised version.

5. We will transfer the definitions from Section 3 to Section 2.

6. We will present our results in a better way as suggested.

7. We will make the point clearer that the economy module in our model is global, i.e., there are no regional economic specifications. Therefore we are unable to provide further information on regions. However, we would like to stress that for our research question, global rather than regional mitigation costs are sufficient. Quite the contrary, for SRM, the side-effect category we investigate here lives

on regional climate pattern discrepancies, for which reason we need to model climate in a regionalized manner. We will use reviewers' comments and explain this point more clearly in the main text and elaborate on how multi-regional model results could differ from ours.

8. We will further ask colleagues and a professional language editor to assure the readability of our manuscript.

9. We will clarify the point that we utilize 'uncertain' in the sense of IPCC AR5 WGIII Annex I: It includes any lack of knowledge; in particular, it also covers probabilistic knowledge (IPCC (2014)).

10. Based on the discussion (including the one cited in our twin paper on SRM Roshan et al. (2019)), we will make the innovative aspect of our MS much clearer. None of the cited articles tackles a joint mitigation / SRM integrated assessment, including regional pattern mismatches induced by SRM. More specifically, Smith and Rasch (2013) focus on the main effect of SRM and answer how much SRM is to be added for a given greenhouse gas emission scenario if a certain global mean temperature target is to be complied with. Ekholm and Korhonen (2016) consider the effect of the timing uncertainty about when SRM can be deployed. Emmerling and Tavoni (2018) investigate the effects of uncertainty about SRM effectiveness and anticipated future learning about it on first-period decisions. Lawrence et al. (2018) present a meta-assessment on whether SRM can play a significant role in compliance with temperature targets at all. None of them discusses a new metric to extend the ethics represented by the $2°C$ target to assess one category of side-effects of SRM.

11. We will cite Kriegler and Bruckner (2004) for clarifying what we mean by 'admissible,' that is, 'in compliance with a certain environmental target or further constraints.'

12. We strive at expressing clearer that we add regional environmental constraints without removing the global 2°C target.

13. By this MS, we present a solution to the conceptual evaluation problem that SRM destroys the correlation between the global indicator 'global mean temperature' and regional climate. This means that for decision-makers who accepted compliance with a global mean temperature target as a guideline for decision-making, this can no longer work as a guideline once SRM is applied. By this article hence, we strive to close this evaluation gap. We ask: 'What are the regional analogs, if not to say 'derivatives' of the global mean temperature interval [0...T*], which would have been acceptable for a proponent of a T*-target? Which regional climate change does correspond to T*?' (Within our case, T*=2°C.) We will make this point clearer in the revised version.

14. We will clarify that 'scaling' refers to 'pattern scaling' (see, e.g., Frieler et al. (2012) and Osborn et al. (2016)). The idea is that regional climate can, in first-order Taylor expansion, be predicted from a few global variables such as CO2-induced temperature change vs. SRM-induced temperature change. Again, we will add the relevant equations.

15. We will state more explicitly that the extra admissible area is not based on any calculation of the impacts that the additional change in precipitation could bring about, and these are used only in an illustrative manner.

16. We will briefly introduce what the Giorgi regions stand for or present.

17. We will describe how our model is a more realistic model than Nordhaus's DICE model and compares it to more complex climate-energy-economy models.

18. We will explain the economic module in our model more precisely in the method section with appropriate equations. Also, we will add that the "splitting approach"

is possible because, from the economic module, we only infer the mitigation costs. As we will explain clearer here, the globally aggregated approach is of sufficient complexity to substitute for spatially resolved economic models.

19. We will clarify that the normalized values of 0 and 1 are not the max and min from the 'observations' between preindustrial and 2C scenarios, but they include the 5% or 10% leeway.

20. We will try to make Figures 3 and 4 more informative by adding vertical grids to the figure and explaining the effects.

21. We will make the point on mitigation costs clearer.

22. We will point out the literature where SRM is discussed from a 'last resort' perspective, under delayed scenarios and tipping points.

23. We will condition our statement more precisely on the regional targets' tolerance levels.

24. We will revise Figure 1 as suggested.

25. Figure 3 describes at a glance the effects of qualitatively different scaling coefficients (or ratios thereof) on regional scenarios and their interaction with various targets. We will explain it more carefully.

26. We will strive at making our points clearer regarding Figure 5.

27. We will include more informative titles for the subfigures for Fugures 3, 4, and 5.

28. We will change our title to reflect the reviewers' comments.

29. We will provide a detailed explanation of the procedure followed for obtaining scaling coefficients.

30. We will present our data as well as AOGCMs more properly and straightforwardly. Regarding the methodology to calculate CSRM and CCo2, we will explain precisely the procedure.

31. We will sharpen the abstract. We indeed want to study the inclusion of one crucial category of side-effect of SRM into an ethical framework coherent with a temperature target. This is our innovation, and its effects can be studied best if applied to the combination SRM+mitigation.

32. We will comply with the journal's data policy and present a proper reference to our data.

**References**

Asseng, S., Foster, I., and Turner, N. C.: The impact of temperature variability on wheat yields, Global Change Biology, 17, 997–1012, 2011.

Ekholm, T. and Korhonen, H.: Climate change mitigation strategy under an uncertain Solar Radiation Management possibility, Climatic Change, 139, 503–515, 2016.

Emmerling, J. and Tavoni, M.: Climate engineering and abatement: A 'flat' relationship under uncertainty, Environmental and resource economics, 69, 395–415, 2018.

Frieler, K., Meinshausen, M., Mengel, M., Braun, N., and Hare, W.: A scaling approach to probabilistic assessment of regional climate change, Journal of Climate, 25, 3117–3144, 2012.

IPCC. Climate change 2014: Mitigation of climate change. Working group III contribution to the fifth assessment report of the intergovernmental panel on climate change, 2014.

Kravitz, B., MacMartin, D. G., Robock, A., Rasch, P. J., Ricke, K. L., Cole, J. N., Curry, C. L., Irvine, P. J., Ji, D., Keith, D. W., et al.: A multi-model assessment of regional climate disparities caused by solar geoengineering, Environmental Research Letters, 9, 074 013, 2014.

Kriegler, E. and Bruckner, T.: Sensitivity analysis of emissions corridors for the 21st century, Climatic change, 66, 345–387, 2004.

Lawrence, M. G., Schäfer, S., Muri, H., Scott, V., Oschlies, A., Vaughan, N. E., Boucher, O., Schmidt, H., Haywood, J., and Scheffran, J.: Evaluating climate geoengineering proposals in the context of the Paris Agreement temperature goals, Nature communications, 9, 1–19, 2018.

Osborn, T. J., Wallace, C. J., Harris, I. C., and Melvin, T. M.: Pattern scaling using ClimGen: monthly-resolution future climate scenarios including changes in the variability of precipitation, Climatic Change, 134, 353–369, 2016.

Portmann, F. T., Siebert, S., and Döll, P.: MIRCA2000—Global monthly irrigated and rainfed crop areas around the year 2000: A new high-resolution data set for agricultural and hydrological modeling, Global biogeochemical cycles, 24, 2010.

Roshan, E., Khabbazan, M. M., and Held, H.: Cost-Risk Trade-Off of Mitigation and Solar Geoengineering: Considering Regional Disparities Under Probabilistic Climate Sensitivity, Environmental and Resource Economics, pp. 1–17, 2019.

Smith, S. J. and Rasch, P. J.: The long-term policy context for solar radiation management, Climatic Change, 121, 487–497, 2013.

---

## Author Response (AR1)

We are very grateful to the reviewers for the concise and constructive reports on our MS *"Trade-offs of Solar Geoengineering and Mitigation under Climate Targets."* We have made a point-by-point reply to the reviewers' comments, which can be found under https://doi.org/10.5194/esd-2020-95-AC1 and https://doi.org/10.5194/esd-2020-95-AC2. Here, we present our Author's Response, which includes our main revisions. Our Author's Response here can be seen as a summary of our detailed responses to the reviewers' comments.

1. We have made the point clearer that while our guardrails are merely hypothetical, they are ethically unique in the sense that they can readily be derived from the global 2°C target. Hence, they provide the straightforward regionalization of that global temperature target.

2. Based on the following two arguments: (i) Due to the climate pattern mismatch between SRM and CO2-induced climate changes, we need to generalize a global target to a set of regional targets, and (ii) Temperature (Asseng et al. (2011)) and precipitation (Portmann et al. (2010)) are highly relevant for agricultural productivity, the pattern mismatch of precipitation is of a larger order of magnitude than that of temperature (Kravitz et al. (2014)). Therefore, we have clarified that temperature and precipitation are the only relevant climate predictors for agricultural productivity or the functionality of ecosystems in general, but we acknowledge precipitation limits as a necessary boundary condition within a target-based framework.

3. We have clarified Section 2 by adding equations for guardrails and a table to show the differences between scenarios. The key point of our ms is to extend the decision-analytic framework of global mean temperature-based decision-making to a situation where global mean temperature ceases being a good predictor of regional climate.

4. We have clarified all of the points about including the level and time profile of SRM, geographical application of SRM, and starting points in the revised version.

5. We have transfered the definitions from Section 3 to Section 2.

6. We have presented our results in a better way as suggested.

7. We have made the point clearer that the economy module in our model is global, i.e., there are no regional economic specifications. Therefore we are unable to provide further information on regions. However, we would like to stress that for our research question, global rather than regional mitigation costs are sufficient. Quite the contrary, for SRM, the side-effect category we investigate here lives on regional climate pattern discrepancies, for which reason we need to model climate in a regionalized manner. We have used reviewers' comments and explained this point more clearly in the main text and elaborated on how multi-regional model results could differ from ours.

8. We have further asked colleagues to assure the readability of our manuscript.

9. We have clarified the point that we utilize 'uncertain' in the sense of IPCC AR5 WGIII Annex I: It includes any lack of knowledge; in particular, it also covers probabilistic knowledge (IPCC (2014)).

10. Based on the discussion (including the one cited in our twin paper on SRM Roshan et al. (2019)), we have made the innovative aspect of our MS much clearer. None of the cited articles tackles a joint mitigation / SRM integrated assessment, including regional pattern mismatches induced by SRM. More specifically, Smith and Rasch (2013) focus on the main effect of SRM and answer how much SRM is to be added for a given greenhouse gas emission scenario if a certain global mean temperature target is to be complied with. Ekholm and Korhonen (2016) consider the effect of the timing uncertainty about when SRM can be deployed. Emmerling and Tavoni (2018) investigate the effects of uncertainty about SRM effectiveness and anticipated future learning about it on first-period decisions. Lawrence et al. (2018) present a meta-assessment on whether SRM can play a significant role in compliance with temperature targets at all. None of them discusses a new metric to extend the ethics represented by the 2°C target to assess one category of side-effects of SRM.

11. We have cited Kriegler and Bruckner (2004) for clarifying what we mean by 'admissible,' that is, 'in compliance with a certain environmental target or further constraints.'

12. We strived at expressing clearer that we add regional environmental constraints without removing the global 2°C target.

13. By this MS, we present a solution to the conceptual evaluation problem that SRM destroys the correlation between the global indicator 'global mean temperature' and regional climate. This means that for decision-makers who accepted compliance with a global mean temperature target as a guideline for decision-making, this can no longer work as a guideline once SRM is applied. By this article hence, we strive to close this evaluation gap. We ask: 'What are the regional analogs, if not to say 'derivatives' of the global mean temperature interval [0...T*], which would have been acceptable for a proponent of a T*-target? Which regional climate change does correspond to T*?' (Within our case, T*=2°C.) We have made this point clearer in the revised version.

14. We have clarified that 'scaling' refers to 'pattern scaling' (see, e.g., Frieler et al. (2012) and Osborn et al. (2016)). The idea is that regional climate can, in first-order Taylor expansion, be predicted from a few global variables such as CO2-induced temperature change vs. SRM-induced temperature change. Again, we have added the relevant equations.

15. We have stated more explicitly that the extra admissible area is not based on any calculation of the impacts that the additional change in precipitation could bring about, and these are used only in an illustrative manner.

16. We have briefly introduced what the Giorgi regions stand for or present.

17. We have described how our model is a more realistic model than Nordhaus's DICE model and compares it to more complex climate-energy-economy models.

18. We have explained the economic module in our model more precisely in the method section. Also, we have added that the "splitting approach" is possible because, from the economic module, we only infer the mitigation costs. As we will explain clearer here, the globally aggregated approach is of sufficient complexity to substitute for spatially resolved economic models.

19. We have clarified that the normalized values of 0 and 1 are not the max and min from the 'observations' between preindustrial and 2C scenarios, but they include the 5% or 10% leeway.

20. We have made the point on mitigation costs clearer.

21. We have pointed out the literature where SRM is discussed from a 'last resort' perspective, under delayed scenarios and tipping points.

22. We have conditioned our statement more precisely on the regional targets' tolerance levels.

23. We have revised Figure 1 as suggested.

24. Figure 3 describes at a glance the effects of qualitatively different scaling coefficients (or ratios thereof) on regional scenarios and their interaction with various targets. We have explained it more carefully.

25. We have strived at making our points clearer regarding Figure 5.

26. We have included more informative titles for the subfigures for Fugures 3, 4, and 5.

27. We have changed our title to reflect the reviewers' comments.

28. We have provided a detailed explanation of the procedure followed for obtaining scaling coefficients.

29. We have presented our data as well as AOGCMs more properly and straightforwardly. Regarding the methodology to calculate CSRM and CCo2, we have explained precisely the procedure.

30. We have sharpened the abstract. We indeed want to study the inclusion of one crucial category of side-effect of SRM into an ethical framework coherent with a temperature target. This is our innovation, and its effects can be studied best if applied to the combination SRM+mitigation.

31. We have complied with the journal's data policy and present a proper reference to our data.

**References**

Asseng, S., Foster, I., and Turner, N. C.: The impact of temperature variability on wheat yields, Global Change Biology, 17, 997–1012, 2011.

Ekholm, T. and Korhonen, H.: Climate change mitigation strategy under an uncertain Solar Radiation Management possibility, Climatic Change, 139, 503–515, 2016.

Emmerling, J. and Tavoni, M.: Climate engineering and abatement: A 'flat' relationship under uncertainty, Environmental and resource economics, 69, 395–415, 2018.

Frieler, K., Meinshausen, M., Mengel, M., Braun, N., and Hare, W.: A scaling approach to probabilistic assessment of regional climate change, Journal of Climate, 25, 3117–3144, 2012.

IPCC. Climate change 2014: Mitigation of climate change. Working group III contribution to the fifth assessment report of the intergovernmental panel on climate change, 2014.

Kravitz, B., MacMartin, D. G., Robock, A., Rasch, P. J., Ricke, K. L., Cole, J. N., Curry, C. L., Irvine, P. J., Ji, D., Keith, D. W., et al.: A multi-model assessment of regional climate disparities caused by solar geoengineering, Environmental Research Letters, 9, 074 013, 2014.

Kriegler, E. and Bruckner, T.: Sensitivity analysis of emissions corridors for the 21st century, Climatic change, 66, 345–387, 2004.

Lawrence, M. G., Schäfer, S., Muri, H., Scott, V., Oschlies, A., Vaughan, N. E., Boucher, O., Schmidt, H., Haywood, J., and Scheffran, J.: Evaluating climate geoengineering proposals in the context of the Paris Agreement temperature goals, Nature communications, 9, 1–19, 2018.

Osborn, T. J., Wallace, C. J., Harris, I. C., and Melvin, T. M.: Pattern scaling using ClimGen: monthly-resolution future climate scenarios including changes in the variability of precipitation, Climatic Change, 134, 353–369, 2016.

Portmann, F. T., Siebert, S., and Döll, P.: MIRCA2000—Global monthly irrigated and rainfed crop areas around the year 2000: A new high-resolution data set for agricultural and hydrological modeling, Global biogeochemical cycles, 24, 2010.

Roshan, E., Khabbazan, M. M., and Held, H.: Cost-Risk Trade-Off of Mitigation and Solar Geoengineering: Considering Regional Disparities Under Probabilistic Climate Sensitivity, Environmental and Resource Economics, pp. 1–17, 2019.

Smith, S. J. and Rasch, P. J.: The long-term policy context for solar radiation management, Climatic Change, 121, 487–497, 2013.